# A closer look at the approximation capabilities of neural networks

**Kai Fong Ernest Chong**
Information Systems Technology and Design (ISTD) pillar,
Singapore University of Technology and Design, Singapore
`ernest_chong@sutd.edu.sg`

## Abstract

The *universal approximation theorem*, in one of its most general versions, says that if we consider only continuous activation functions $\sigma$, then a standard feedforward neural network with one hidden layer is able to approximate any continuous multivariate function $f$ to any given approximation threshold $\varepsilon$, if and only if $\sigma$ is non-polynomial. In this paper, we give a direct algebraic proof of the theorem. Furthermore we shall explicitly quantify the number of hidden units required for approximation. Specifically, if $X \subseteq \mathbb{R}^n$ is compact, then a neural network with $n$ input units, $m$ output units, and a single hidden layer with $\binom{n+d}{d}$ hidden units (independent of $m$ and $\varepsilon$), can uniformly approximate any polynomial function $f : X \to \mathbb{R}^m$ whose total degree is at most $d$ for each of its $m$ coordinate functions. In the general case that $f$ is any continuous function, we show there exists some $N \in \mathcal{O}(\varepsilon^{-n})$ (independent of $m$), such that $N$ hidden units would suffice to approximate $f$. We also show that this uniform approximation property (UAP) still holds even under seemingly strong conditions imposed on the weights. We highlight several consequences: (i) For any $\delta > 0$, the UAP still holds if we restrict all non-bias weights $w$ in the last layer to satisfy $|w| < \delta$. (ii) There exists some $\lambda > 0$ (depending only on $f$ and $\sigma$), such that the UAP still holds if we restrict all non-bias weights $w$ in the first layer to satisfy $|w| > \lambda$. (iii) If the non-bias weights in the first layer are *fixed* and randomly chosen from a suitable range, then the UAP holds with probability 1.

## 1    Introduction and Overview

A standard (feedforward) neural network with $n$ input units, $m$ output units, and with one or more hidden layers, refers to a computational model $\mathcal{N}$ that can compute a certain class of functions $\rho : \mathbb{R}^n \to \mathbb{R}^m$, where $\rho = \rho_W$ is parametrized by $W$ (called the *weights* of $\mathcal{N}$). Implicitly, the definition of $\rho$ depends on a choice of some fixed function $\sigma : \mathbb{R} \to \mathbb{R}$, called the *activation function* of $\mathcal{N}$. Typically, $\sigma$ is assumed to be continuous, and historically, the earliest commonly used activation functions were sigmoidal.

A key fundamental result justifying the use of sigmoidal activation functions was due to Cybenko (1989), Hornik et al. (1989), and Funahashi (1989), who independently proved the first version of what is now famously called the *universal approximation theorem*. This first version says that if $\sigma$ is sigmoidal, then a standard neural network with one hidden layer would be able to uniformly approximate any continuous function $f : X \to \mathbb{R}^m$ whose domain $X \subseteq \mathbb{R}^n$ is compact. Hornik (1991) extended the theorem to the case when $\sigma$ is any continuous bounded non-constant activation function. Subsequently, Leshno et al. (1993) proved that for the class of continuous activation functions, a standard neural network with one hidden layer is able to uniformly approximate any continuous function $f : X \to \mathbb{R}^m$ on any compact $X \subseteq \mathbb{R}^n$, if and only if $\sigma$ is non-polynomial.

Although a single hidden layer is sufficient for the uniform approximation property (UAP) to hold, the number of hidden units required could be arbitrarily large. Given a subclass $\mathcal{F}$ of real-valued continuous functions on a compact set $X \subseteq \mathbb{R}^n$, a fixed activation function $\sigma$, and some $\varepsilon > 0$, let $N = N(\mathcal{F}, \sigma, \varepsilon)$ be the minimum number of hidden units required for a single-hidden-layer neural network to be able to uniformly approximate *every* $f \in \mathcal{F}$ within an approximation error

threshold of $\varepsilon$. If $\sigma$ is the rectified linear unit (ReLU) $x \mapsto \max(0, x)$, then $N$ is at least $\Omega(\frac{1}{\sqrt{\varepsilon}})$ when $\mathcal{F}$ is the class of $C^2$ non-linear functions (Yarotsky, 2017), or the class of strongly convex differentiable functions (Liang & Srikant, 2016); see also (Arora et al., 2018). If $\sigma$ is any smooth non-polynomial function, then $N$ is at most $\mathcal{O}(\varepsilon^{-n})$ for the class of $C^1$ functions with bounded Sobolev norm (Mhaskar, 1996); cf. (Pinkus, 1999, Thm. 6.8), (Maiorov & Pinkus, 1999). As a key highlight of this paper, we show that if $\sigma$ is an *arbitrary* continuous non-polynomial function, then $N$ is at most $\mathcal{O}(\varepsilon^{-n})$ for the entire class of continuous functions. In fact, we give an explicit upper bound for $N$ in terms of $\varepsilon$ and the modulus of continuity of $f$, so better bounds could be obtained for certain subclasses $\mathcal{F}$, which we discuss further in Section 4. Furthermore, even for the wider class $\mathcal{F}$ of all continuous functions $f : X \to \mathbb{R}^m$, the bound is still $\mathcal{O}(\varepsilon^{-n})$, independent of $m$.

To prove this bound, we shall give a direct algebraic proof of the universal approximation theorem, in its general version as stated by Leshno et al. (1993) (i.e. $\sigma$ is continuous and non-polynomial). An important advantage of our algebraic approach is that we are able to glean additional information on sufficient conditions that would imply the UAP. Another key highlight we have is that if $\mathcal{F}$ is the subclass of polynomial functions $f : X \to \mathbb{R}^m$ with total degree at most $d$ for each coordinate function, then $\binom{n+d}{d}$ hidden units would suffice. In particular, notice that our bound $N \le \binom{n+d}{d}$ does not depend on the approximation error threshold $\varepsilon$ or the output dimension $m$.

We shall also show that the UAP holds even under strong conditions on the weights. Given any $\delta > 0$, we can always choose the non-bias weights in the last layer to have small magnitudes no larger than $\delta$. Furthermore, we show that there exists some $\lambda > 0$ (depending only on $\sigma$ and the function $f$ to be approximated), such that the non-bias weights in the first layer can always be chosen to have magnitudes greater than $\lambda$. Even with these seemingly strong restrictions on the weights, we show that the UAP still holds. Thus, our main results can be collectively interpreted as a quantitative refinement of the universal approximation theorem, with extensions to restricted weight values.

**Outline:** Section 2 covers the preliminaries, including relevant details on arguments involving dense sets. Section 3 gives precise statements of our results, while Section 4 discusses the consequences of our results. Section 5 introduces our algebraic approach and includes most details of the proofs of our results; details omitted from Section 5 can be found in the appendix. Finally, Section 6 concludes our paper with further remarks.

## 2 PRELIMINARIES

### 2.1 NOTATION AND DEFINITIONS

Let $\mathbb{N}$ be the set of non-negative integers, let $\mathbf{0}_n$ be the zero vector in $\mathbb{R}^n$, and let $\mathrm{Mat}(k, \ell)$ be the vector space of all $k$-by-$\ell$ matrices with real entries. For any function $f : \mathbb{R}^n \to \mathbb{R}^m$, let $f^{[t]}$ denote the $t$-th coordinate function of $f$ (for each $1 \le t \le m$). Given $\alpha = (\alpha_1, \ldots, \alpha_n) \in \mathbb{N}^n$ and any $n$-tuple $x = (x_1, \ldots, x_n)$, we write $x^\alpha$ to mean $x_1^{\alpha_1} \cdots x_n^{\alpha_n}$. If $x \in \mathbb{R}^n$, then $x^\alpha$ is a real number, while if $x$ is a sequence of variables, then $x^\alpha$ is a *monomial*, i.e. an $n$-variate polynomial with a single term. Let $\mathcal{W}_N^{n,m} := \{W \in \mathrm{Mat}(n + 1, N) \times \mathrm{Mat}(N + 1, m)\}$ for each $N \ge 1$, and define $\mathcal{W}^{n,m} = \bigcup_{N \ge 1} \mathcal{W}_N^{n,m}$. If the context is clear, we supress the superscripts $n, m$ in $\mathcal{W}_N^{n,m}$ and $\mathcal{W}^{n,m}$.

Given any $X \subseteq \mathbb{R}^n$, let $\mathcal{C}(X)$ be the vector space of all continuous functions $f : X \to \mathbb{R}$. We use the convention that every $f \in \mathcal{C}(X)$ is a function $f(x_1, \ldots, x_n)$ in terms of the variables $x_1, \ldots, x_n$, unless $n = 1$, in which case $f$ is in terms of a single variable $x$ (or $y$). We say $f$ is *non-zero* if $f$ is not identically the zero function on $X$. Let $\mathcal{P}(X)$ be the subspace of all polynomial functions in $\mathcal{C}(X)$. For each $d \in \mathbb{N}$, let $\mathcal{P}_{\le d}(X)$ (resp. $\mathcal{P}_d(X)$) be the subspace consisting of all polynomial functions of total degree $\le d$ (resp. exactly $d$). More generally, let $\mathcal{C}(X, \mathbb{R}^m)$ be the vector space of all continuous functions $f : X \to \mathbb{R}^m$, and define $\mathcal{P}(X, \mathbb{R}^m), \mathcal{P}_{\le d}(X, \mathbb{R}^m), \mathcal{P}_d(X, \mathbb{R}^m)$ analogously.

Throughout, we assume that $\sigma \in \mathcal{C}(\mathbb{R})$. For every $W = (W^{(1)}, W^{(2)}) \in \mathcal{W}$, let $\mathbf{w}_j^{(k)}$ be the $j$-th column vector of $W^{(k)}$, and let $w_{i,j}^{(k)}$ be the $(i, j)$-th entry of $W^{(k)}$ (for $k = 1, 2$). The index $i$ begins at $i = 0$, while the indices $j, k$ begin at $j = 1, k = 1$ respectively. For convenience, let $\widehat{\mathbf{w}}_j^{(k)}$ denote the truncation of $\mathbf{w}_j^{(k)}$ obtained by removing the first entry $w_{0,j}^{(k)}$. Define the function $\rho_W^\sigma : \mathbb{R}^n \to \mathbb{R}^m$ so that for each $1 \le j \le m$, the $j$-th coordinate function $\rho_W^{\sigma[j]}$ is given by the map

$$x \mapsto w_{0,j}^{(2)} + \sum_{i=1}^N w_{i,j}^{(2)} \sigma(\mathbf{w}_i^{(1)} \cdot (1, x)),$$

where "·" denotes dot product, and $(1, x)$ denotes a column vector in $\mathbb{R}^{n+1}$ formed by concatenating 1 before $x$. The class of functions that neural networks $\mathcal{N}$ with one hidden layer can compute is precisely $\{\rho_W^\sigma : W \in \mathcal{W}\}$, where $\sigma$ is called the *activation function* of $\mathcal{N}$ (or of $\rho_W^\sigma$). Functions $\rho_W^\sigma$ satisfying $W \in \mathcal{W}_N$ correspond to neural networks with $N$ hidden units (in its single hidden layer). Every $w_{i,j}^{(k)}$ is called a *weight* in the $k$-th layer, where $w_{i,j}^{[k]}$ is called a *bias weight* (resp. *non-bias weight*) if $i = 0$ (resp. $i \neq 0$).

Notice that we do not apply the activation function $\sigma$ to the output layer. This is consistent with previous approximation results for neural networks. The reason is simple: $\sigma \circ \rho_W^{\sigma\,[j]}$ (restricted to domain $X \subseteq \mathbb{R}^n$) cannot possibly approximate $f : X \to \mathbb{R}$ if there exists some $x_0 \in X$ such that $\sigma(X)$ is bounded away from $f(x_0)$. If instead $f(X)$ is contained in the closure of $\sigma(X)$, then applying $\sigma$ to $\rho_W^{\sigma\,[j]}$ has essentially the same effect as allowing for bias weights $w_{0,j}^{(2)}$.

Although some authors, e.g. (Leshno et al., 1993), do not explicitly include bias weights in the output layer, the reader should check that if $\sigma$ is not identically zero, say $\sigma(y_0) \neq 0$, then having a bias weight $w_{0,j}^{(2)} = c$ is equivalent to setting $w_{0,j}^{(2)} = 0$ (i.e. no bias weight in the output layer) and introducing an $(N + 1)$-th hidden unit, with corresponding weights $w_{0,N+1}^{(1)} = y_0$, $w_{i,N+1}^{(1)} = 0$ for all $1 \leq i \leq n$, and $w_{N+1,j}^{(2)} = \frac{c}{\sigma(y_0)}$; this means our results also apply to neural networks without bias weights in the output layer (but with one additional hidden unit).

## 2.2 Arguments involving dense subsets

A key theme in this paper is the use of dense subsets of metric spaces. We shall consider several notions of "dense". First, recall that a *metric* on a set $S$ is any function $\mathfrak{d} : S \times S \to \mathbb{R}$ such that for all $x, y, z \in S$, the following conditions hold:
   (i) $\mathfrak{d}(x, y) \geq 0$, with equality holding if and only if $x = y$;
   (ii) $\mathfrak{d}(x, y) = \mathfrak{d}(y, x)$;
   (iii) $\mathfrak{d}(x, z) \leq \mathfrak{d}(x, y) + \mathfrak{d}(y, z)$.

The set $S$, together with a metric on $S$, is called a *metric space*. For example, the usual Euclidean norm for vectors in $\mathbb{R}^n$ gives the *Euclidean metric* $(u, v) \mapsto \|u - v\|_2$, hence $\mathbb{R}^n$ is a metric space. In particular, every pair in $\mathcal{W}_N$ can be identified with a vector in $\mathbb{R}^{(m+n+1)N}$, so $\mathcal{W}_N$, together with the Euclidean metric, is a metric space.

Given a metric space $X$ (with metric $\mathfrak{d}$), and some subset $U \subseteq X$, we say that $U$ is *dense* in $X$ (w.r.t. $\mathfrak{d}$) if for all $\varepsilon > 0$ and all $x \in X$, there exists some $u \in U$ such that $\mathfrak{d}(x, u) < \varepsilon$. Arbitrary unions of dense subsets are dense. If $U \subseteq U' \subseteq X$ and $U$ is dense in $X$, then $U'$ must also be dense in $X$.

A basic result in algebraic geometry says that if $p \in \mathcal{P}(\mathbb{R}^n)$ is non-zero, then $\{x \in \mathbb{R}^n : p(x) \neq 0\}$ is a dense subset of $\mathbb{R}^n$ (w.r.t. the Euclidean metric). This subset is in fact an open set in the Zariski topology, hence any finite intersection of such *Zariski-dense open sets* is dense; see (Eisenbud, 1995). More generally, the following is true: Let $p_1, \ldots, p_k \in \mathcal{P}(\mathbb{R}^n)$, and suppose that $X := \{x \in \mathbb{R}^n : p_i(x) = 0 \text{ for all } 1 \leq i \leq k\}$. If $p \in \mathcal{P}(X)$ is non-zero, then $\{x \in X : p(x) \neq 0\}$ is a dense subset of $X$ (w.r.t. the Euclidean metric). In subsequent sections, we shall frequently use these facts.

Let $X \subseteq \mathbb{R}^n$ be a compact set. (Recall that $X$ is *compact* if it is bounded and contains all of its limit points.) For any real-valued function $f$ whose domain contains $X$, the *uniform norm* of $f$ on $X$ is $\|f\|_{\infty,X} := \sup\{|f(x)| : x \in X\}$. More generally, if $f : X \to \mathbb{R}^m$, then we define $\|f\|_{\infty,X} := \max\{\|f^{[j]}\|_{\infty,X} : 1 \leq j \leq m\}$. The uniform norm of functions on $X$ gives the *uniform metric* $(f, g) \mapsto \|f - g\|_{\infty,X}$, hence $\mathcal{C}(X)$ is a metric space.

## 2.3 Background on approximation theory

**Theorem 2.1** (Stone–Weirstrass theorem). *Let $X \subseteq \mathbb{R}^n$ be compact. For any $f \in \mathcal{C}(X)$, there exists a sequence $\{p_k\}_{k \in \mathbb{N}}$ of polynomial functions in $\mathcal{P}(X)$ such that $\lim_{k \to \infty} \|f - p_k\|_{\infty,X} = 0$.*

Let $X \subseteq \mathbb{R}$ be compact. For all $d \in \mathbb{N}$ and $f \in \mathcal{C}(X)$, define

$$E_d(f) := \inf\{\|f - p\|_{\infty,X} : p \in \mathcal{P}_{\leq d}(X)\}. \tag{1}$$

A central result in approximation theory, due to Chebyshev, says that for fixed $d, f$, the infimum in (1) is attained by some unique $p^* \in \mathcal{P}_{\leq d}(\mathbb{R})$; see (Rivlin, 1981, Chap. 1). (Notice here that we define $p^*$ to have domain $\mathbb{R}$.) This unique polynomial $p^*$ is called the *best polynomial approximant* to $f$ of degree $d$.

Given a metric space $X$ with metric $\mathfrak{d}$, and any uniformly continuous function $f : X \to \mathbb{R}$, the *modulus of continuity* of $f$ is a function $\omega_f : [0, \infty] \to [0, \infty]$ defined by

$$\omega_f(\delta) := \sup\{|f(x) - f(y)| : x, y \in X, \mathfrak{d}(x, y) \leq \delta\}.$$

By the Heine–Cantor theorem, any continuous $f$ with a compact domain is uniformly continuous.

**Theorem 2.2** (Jackson's theorem; see (Rivlin, 1981, Cor. 1.4.1))**.** *Let $d \geq 1$ be an integer, and let $Y \subseteq \mathbb{R}$ be a closed interval of length $r \geq 0$. Suppose $f \in \mathcal{C}(Y)$, and let $p^*$ be the best polynomial approximant to $f$ of degree $d$. Then $\|f - p^*\|_{\infty, Y} = E_d(f) \leq 6\omega_f(\frac{r}{2d})$.*

## 3    MAIN RESULTS

Throughout this section, let $X \subseteq \mathbb{R}^n$ be a compact set.

**Theorem 3.1.** *Let $d \geq 2$ be an integer, and let $f \in \mathcal{P}_{\leq d}(X, \mathbb{R}^m)$ (i.e. each coordinate function $f^{[t]}$ has total degree $\leq d$). If $\sigma \in \mathcal{C}(\mathbb{R}) \backslash \mathcal{P}_{\leq d-1}(\mathbb{R})$, then for every $\varepsilon > 0$, there exists some $W \in \mathcal{W}_{\binom{n+d}{d}}$ such that $\|f - \rho_W^\sigma\|_{\infty, X} < \varepsilon$. Furthermore, the following holds:*

*(i) Given any $\lambda > 0$, we can choose this $W$ to satisfy the condition that $|w_{i,j}^{(2)}| < \lambda$ for all non-bias weights $w_{i,j}^{(2)}$ (i.e. $i \neq 0$) in the second layer.*

*(ii) There exists some $\lambda' > 0$, depending only on $f$ and $\sigma$, such that we could choose the weights of $W$ in the first layer to satisfy the condition that $\|\widehat{\mathbf{w}}_j^{(1)}\|_2 > \lambda'$ for all $j$.*

**Theorem 3.2.** *Let $f \in \mathcal{C}(X, \mathbb{R}^m)$, and suppose $\sigma \in \mathcal{C}(\mathbb{R}) \backslash \mathcal{P}(\mathbb{R})$. Then for every $\varepsilon > 0$, there exists an integer $N \in \mathcal{O}(\varepsilon^{-n})$ (independent of $m$), and some $W \in \mathcal{W}_N$, such that $\|f - \rho_W^\sigma\|_{\infty, X} < \varepsilon$. In particular, if we let $D := \sup\{\|x - y\|_2 : x, y \in X\}$ be the diameter of $X$, then we can set $N = \binom{n+d_\varepsilon}{d_\varepsilon}$, where $d_\varepsilon := \min\{d \in \mathbb{Z} : d \geq 2, \omega_{f^{[t]}}(\frac{D}{2d}) < \frac{\varepsilon}{6}$ for all $1 \leq t \leq m\}$. (Note that $d_\varepsilon$ is well-defined, since $\lim_{\delta \to 0^+} \omega_{f^{[t]}}(\delta) = 0$ for each $t$.) Furthermore, we could choose this $W$ to satisfy either (i) or (ii), where (i), (ii) are conditions on $W$ as described in Theorem 3.1.*

**Theorem 3.3.** *Let $f \in \mathcal{C}(X, \mathbb{R}^m)$, and suppose that $\sigma \in \mathcal{C}(\mathbb{R}) \backslash \mathcal{P}(\mathbb{R})$. Then there exists $\lambda > 0$ (which depends only on $f$ and $\sigma$) such that for every $\varepsilon > 0$, there exists an integer $N$ (independent of $m$) such that the following holds:*

*Let $W \in \mathcal{W}_N$ such that each $\widehat{\mathbf{w}}_j^{(1)} \in \mathbb{R}^n$ (for $1 \leq j \leq N$) is chosen uniformly at random from the set $\{\mathbf{u} \in \mathbb{R}^n : \|\mathbf{u}\|_2 > \lambda\}$. Then, with probability 1, there exist choices for the bias weights $w_{0,j}^{(1)}$ (for $1 \leq j \leq N$) in the first layer, and (both bias and non-bias) weights $w_{i,j}^{(2)}$ in the second layer, such that $\|f - \rho_W^\sigma\|_{\infty, X} < \varepsilon$.*

*Moreover, $N \in \mathcal{O}(\varepsilon^{-n})$ for general $f \in \mathcal{C}(X, \mathbb{R}^m)$, and we can let $N = \binom{n+d}{d}$ if $f \in \mathcal{P}_{\leq d}(X, \mathbb{R}^m)$.*

## 4    DISCUSSION

The universal approximation theorem (version of Leshno et al. (1993)) is an immediate consequence of Theorem 3.2 and the observation that $\sigma$ must be non-polynomial for the UAP to hold, which follows from the fact that the uniform closure of $\mathcal{P}_{\leq d}(X)$ is $\mathcal{P}_{\leq d}(X)$ itself, for every integer $d \geq 1$. Alternatively, we could infer the universal approximation theorem by applying the Stone–Weirstrass theorem (Theorem 2.1) to Theorem 3.1.

Given fixed $n, m, d$, a compact set $X \subseteq \mathbb{R}^n$, and $\sigma \in \mathcal{C}(\mathbb{R}) \backslash \mathcal{P}_{\leq d-1}(\mathbb{R})$, Theorem 3.1 says that we could use a *fixed* number $N$ of hidden units (independent of $\varepsilon$) and still be able to approximate any function $f \in \mathcal{P}_{\leq d}(X, \mathbb{R}^m)$ to any desired approximation error threshold $\varepsilon$. Our $\varepsilon$-free bound, although possibly surprising to some readers, is not the first instance of an $\varepsilon$-free bound: Neural networks with *two* hidden layers of sizes $2n + 1$ and $4n + 3$ respectively are able to uniformly approximate any $f \in \mathcal{C}(X)$, provided that we use a (somewhat pathological) activation function (Maiorov

& Pinkus, 1999); cf. (Pinkus, 1999). Lin et al. (2017) showed that for fixed $n, d$, and a fixed smooth non-linear $\sigma$, there is a fixed $N$ (i.e. $\varepsilon$-free), such that a neural network with $N$ hidden units is able to approximate any $f \in \mathcal{P}_{\leq d}(X)$. An explicit expression for $N$ is not given, but we were able to infer from their constructive proof that $N = 4\binom{n+d+1}{d} - 4$ hidden units are required, over $d - 1$ hidden layers (for $d \geq 2$). In comparison, we require less hidden units and a single hidden layer.

Our proof of Theorem 3.2 is an application of Jackson's theorem (Theorem 2.2) to Theorem 3.1, which gives an explicit bound in terms of the values of the modulus of continuity $\omega_f$ of the function $f$ to be approximated. The moduli of continuity of several classes of continuous functions have explicit characterizations. For example, given constants $k > 0$ and $0 < \alpha \leq 1$, recall that a continuous function $f : X \to \mathbb{R}$ (for compact $X \subseteq \mathbb{R}^n$) is called $k$-*Lipschitz* if $|f(x) - f(y)| \leq k\|x - y\|$ for all $x, y \in X$, and it is called $\alpha$-Hölder if there is some constant $c$ such that $|f(x) - f(y)| \leq c\|x - y\|^\alpha$ for all $x, y \in X$. The modulus of continuity of a $k$-Lipschitz (resp. $\alpha$-Hölder) continuous function $f$ is $\omega_f(t) = kt$ (resp. $\omega_f(t) = ct^\alpha$), hence Theorem 3.2 implies the following corollary.

**Corollary 4.1.** *Suppse $\sigma \in \mathcal{C}(\mathbb{R})\backslash\mathcal{P}(\mathbb{R})$.*
   *(i) If $f : [0,1]^n \to \mathbb{R}$ is $k$-Lipschitz continuous, then for every $\varepsilon > 0$, there exists some $W \in \mathcal{W}_N$ that satisfies $\|f - \rho_W^\sigma\|_{\infty,X} < \varepsilon$, where $N = \binom{n + \lceil \frac{3k}{\varepsilon} \rceil}{n}$.*
   *(ii) If $f : [0,1]^n \to \mathbb{R}$ is $\alpha$-Hölder continuous, then there is a constant $k$ such that for every $\varepsilon > 0$, there exists some $W \in \mathcal{W}_N$ that satisfies $\|f - \rho_W^\sigma\|_{\infty,X} < \varepsilon$, where $N = \binom{n+d}{d}$, and $d = \lceil \frac{1}{2}(\frac{k}{\varepsilon})^{1/\alpha} \rceil$.*

An interesting consequence of Theorem 3.3 is the following: The freezing of lower layers of a neural network, even in the extreme case that all frozen layers are randomly initialized and the last layer is the only "non-frozen" layer, does not necessarily reduce the representability of the resulting model. Specifically, in the single-hidden-layer case, we have shown that if the non-bias weights in the first layer are *fixed* and randomly chosen from some suitable fixed range, then the UAP holds with probability 1, provided that there are sufficiently many hidden units. Of course, this representability does not reveal anything about the learnability of such a model. In practice, layers are already pre-trained before being frozen. It would be interesting to understand quantitatively the difference between having pre-trained frozen layers and having randomly initialized frozen layers.

Theorem 3.3 can be viewed as a result on random features, which were formally studied in relation to kernel methods (Rahimi & Recht, 2007). In the case of ReLU activation functions, Sun et al. (2019) proved an analog of Theorem 3.3 for the approximation of functions in a reproducing kernel Hilbert space; cf. (Rahimi & Recht, 2008). For a good discussion on the role of random features in the representability of neural networks, see (Yehudai & Shamir, 2019).

The UAP is also studied in other contexts, most notably in relation to the depth and width of neural networks. Lu et al. (2017) proved the UAP for neural networks with hidden layers of bounded width, under the assumption that ReLU is used as the activation function. Soon after, Hanin (2017) strengthened the bounded-width UAP result by considering the approximation of continuous convex functions. Recently, the role of depth in the expressive power of neural networks has gathered much interest (Delalleau & Bengio, 2011; Eldan & Shamir, 2016; Mhaskar et al., 2017; Montúfar et al., 2014; Telgarsky, 2016). We do not address depth in this paper, but we believe it is possible that our results can be applied iteratively to deeper neural networks, perhaps in particular for the approximation of compositional functions; cf. (Mhaskar et al., 2017).

## 5 AN ALGEBRAIC APPROACH FOR PROVING UAP

We begin with a "warm-up" result. Subsequent results, even if they seem complicated, are actually multivariate extensions of this "warm-up" result, using very similar ideas.

**Theorem 5.1.** *Let $p(x)$ be a real polynomial of degree $d$ with all-non-zero coefficients, and let $a_1, \ldots, a_{d+1}$ be real numbers. For each $1 \leq j \leq d+1$, define $f_j : \mathbb{R} \to \mathbb{R}$ by $x \mapsto p(a_j x)$. Then $f_1, \ldots, f_{d+1}$ are linearly independent if and only if $a_1, \ldots, a_{d+1}$ are distinct.*

*Proof.* For each $0 \leq i, k \leq d$ and each $1 \leq j \leq d+1$, let $f_j^{(i)}$ (resp. $p^{(i)}$) be the $i$-th derivative of $f_j$ (resp. $p$), and let $\alpha_k^{(i)}$ be the coefficient of $x^k$ in $p^{(i)}(x)$. Recall that the *Wronskian* of $(f_1, \ldots, f_{d+1})$ is defined to be the determinant of the matrix $M(x) := [f_j^{(i-1)}(x)]_{1 \leq i,j \leq d+1}$. Since $f_1, \ldots, f_{d+1}$ are

polynomial functions, it follows that $(f_1, \ldots, f_{d+1})$ is a sequence of linearly independent functions if and only if its Wronskian is not the zero function (LeVeque, 1956, Thm. 4.7(a)). Clearly, if $a_i = a_j$ for $i \neq j$, then $\det M(x)$ is identically zero. Thus, it suffices to show that if $a_1, \ldots, a_{d+1}$ are distinct, then the evaluation $\det M(1)$ of this Wronskian at $x = 1$ gives a non-zero value.

Now, the $(i, j)$-th entry of $M(1)$ equals $a_j^{i-1} p^{(i-1)}(a_j)$, so $M(1) = M' M''$, where $M'$ is an upper triangular matrix whose $(i, j)$-th entry equals $\alpha_{j-i}^{(i-1)}$, and $M'' = [a_j^{i-1}]_{1 \leq i, j \leq d+1}$ is the transpose of a Vandermonde matrix, whose determinant is

$$\det(M'') = \prod_{1 \leq i < j \leq d+1} (a_j - a_i).$$

Note that the $k$-th diagonal entry of $M'$ is $\alpha_0^{(k-1)} = (k-1)! \alpha_{k-1}^{(0)}$, which is non-zero by assumption, so $\det(M') \neq 0$. Thus, if $a_1, \ldots, a_{d+1}$ are distinct, then $\det M(1) = \det(M') \det(M'') \neq 0$. $\quad\square$

**Definition 5.2.** Given $N \geq 1$, $W \in \mathcal{W}_N^{n,m}$, $x_0 \in \mathbb{R}^n$, and any function $g : \mathbb{R} \to \mathbb{R}$, let $\mathcal{F}_{g,x_0}(W)$ denote the sequence of functions $(f_1, \ldots, f_N)$, such that each $f_j : \mathbb{R}^n \to \mathbb{R}$ is defined by the map $x \mapsto g(\widehat{\mathbf{w}}_j^{(1)} \cdot (x - x_0))$. Also, define the set

$$^g \mathcal{W}_{n,N;x_0}^{\text{ind}} := \{W \in \mathcal{W}_N^{n,m} : \mathcal{F}_{g,x_0}(W) \text{ is linearly independent}\}.$$

Note that the value of $m$ is irrelevant for defining $^g \mathcal{W}_{n,N;x_0}^{\text{ind}}$.

**Remark 5.3.** Given $\mathbf{a} = (a_1, \ldots, a_n) \in \mathbb{R}^n$, consider the ring automorphism $\varphi : \mathcal{P}(\mathbb{R}^n) \to \mathcal{P}(\mathbb{R}^n)$ induced by $x_i \mapsto x_i - a_i$ for all $1 \leq i \leq n$. For any $f_1, \ldots, f_k \in \mathcal{P}(\mathbb{R}^n)$ and scalars $\alpha_1, \ldots, \alpha_k \in \mathbb{R}$, note that $\alpha_1 f_1 + \cdots + \alpha_k f_k = 0$ if and only if $\alpha_1 \varphi(f_1) + \cdots + \alpha_k \varphi(f_k) = 0$, thus linear independence is preserved under $\varphi$. Consequently, if the function $g$ in Definition 5.2 is polynomial, then $^g \mathcal{W}_{n,N;x_0}^{\text{ind}} = {}^g \mathcal{W}_{n,N;\mathbf{0}_n}^{\text{ind}}$ for all $x_0 \in \mathbb{R}^n$.

**Corollary 5.4.** *Let $m$ be arbitrary. If $p \in \mathcal{P}_d(\mathbb{R})$ has all-non-zero coefficients, then $^p \mathcal{W}_{1,d+1;0}^{\text{ind}}$ is a dense subset of $\mathcal{W}_{d+1}^{1,m}$ (in the Euclidean metric).*

*Proof.* For all $1 \leq j < j' \leq N$, let $\mathcal{A}_{j,j'} := \{W \in \mathcal{W}_{d+1}^{1,m} : w_{1,j'}^{(1)} - w_{1,j}^{(1)} \neq 0\}$, and note that $\mathcal{A}_{j,j'}$ is dense in $\mathcal{W}_{d+1}^{1,m}$. So by Theorem 5.1, $^p \mathcal{W}_{1,d+1;0}^{\text{ind}} = \bigcap_{1 \leq j < j' \leq N} \mathcal{A}_{j,j'}$ is also dense in $\mathcal{W}_{d+1}^{1,m}$. $\quad\square$

As we have seen in the proof of Theorem 5.1, Vandermonde matrices play an important role. To extend this theorem (and Corollary 5.4) to the multivariate case, we need a generalization of the Vandermonde matrix as described in (D'Andrea & Tabera, 2009). (Other generalizations of the Vandermonde matrix exist in the literature.) First, define the sets

$$\Lambda_{\leq d}^n := \{(\alpha_1, \ldots, \alpha_n) \in \mathbb{N}^n : \alpha_1 + \cdots + \alpha_n \leq d\};$$
$$\mathcal{M}_{\leq d}^n := \{(x \mapsto x^\alpha) \in \mathcal{P}(\mathbb{R}^n) : \alpha \in \Lambda_{\leq d}^n\}.$$

It is easy to show that $|\Lambda_{\leq d}^n| = \binom{n+d}{d}$, and that the set $\mathcal{M}_{\leq d}^n$ of monomial functions forms a basis for $\mathcal{P}_{\leq d}(\mathbb{R}^n)$. Sort the $n$-tuples in $\Lambda_{\leq d}^n$ in colexicographic order, i.e. $(\alpha_1, \ldots, \alpha_n) < (\alpha_1', \ldots, \alpha_n')$ if and only if $\alpha_i < \alpha_i'$ for the largest index $i$ such that $\alpha_i \neq \alpha_i'$. Let $\lambda_1 < \cdots < \lambda_{\binom{n+d}{d}}$ denote all the $\binom{n+d}{d}$ $n$-tuples in $\Lambda_{\leq d}^n$ after sorting. Analogously, let $q_1, \ldots q_{\binom{n+d}{d}}$ be all the monomial functions in $\mathcal{M}_{\leq d}^n$ in this order, i.e. each $q_k : \mathbb{R}^n \to \mathbb{R}$ is given by the map $x \mapsto x^{\lambda_k}$. Given any sequence $(v_1, \ldots, v_{\binom{n+d}{d}})$ of vectors in $\mathbb{R}^n$, the *generalized Vandermonde matrix* associated to it is

$$Q = Q[v_1, \ldots, v_{\binom{n+d}{d}}] := [q_i(v_j)]_{1 \leq i, j \leq \binom{n+d}{d}}. \tag{2}$$

**Definition 5.5.** Given any $W \in \mathcal{W}_{\binom{n+d}{d}}^{n,m}$, we define the *non-bias Vandermonde matrix* of $W$ to be the generalized Vandermonde matrix $Q[W] := [q_i(\widehat{\mathbf{w}}_j^{(1)})]_{1 \leq i, j \leq \binom{n+d}{d}}$ associated to $(\widehat{\mathbf{w}}_1^{(1)}, \ldots, \widehat{\mathbf{w}}_{\binom{n+d}{d}}^{(1)})$.

**Theorem 5.6.** *Let $m$ be arbitrary, let $p \in \mathcal{P}_d(\mathbb{R}^n)$, and suppose that $p$ has all-non-zero coefficients. Also, suppose that $p_1, \ldots, p_k \in \mathcal{P}(\mathcal{W}_{\binom{n+d}{d}}^{n,m})$ are fixed polynomial functions on the non-bias weights of the first layer. Define the following sets:*

$$\mathcal{U} := \{W \in \mathcal{W}_{\binom{n+d}{d}}^{n,m} : p_i(W) = 0 \text{ for all } 1 \leq i \leq k\};$$
$$^p \mathcal{U}^{\text{ind}} := \{W \in \mathcal{U} : \mathcal{F}_{p,\mathbf{0}_n}(W) \text{ is linearly independent}\}.$$

*If there exists $W \in \mathcal{U}$ such that the non-bias Vandermonde matrix of $W$ is non-singular, then $^p \mathcal{U}^{\text{ind}}$ is dense in $\mathcal{U}$ (w.r.t. the Euclidean metric).*

*Proof.* We essentially extend the ideas in the proofs of Theorem 5.1 and Corollary 5.4, using the notion of generalized Wronskians; see Appendix A.1 for proof details. □

**Corollary 5.7.** *Let $m$ be arbitrary. If $p \in \mathcal{P}(\mathbb{R})$ is a fixed polynomial function of degree $d$ with all-non-zero coefficients, then ${}^P\mathcal{W}^{\mathrm{ind}}_{n,\binom{n+d}{d};\mathbf{0}_n}$ is a dense subset of $\mathcal{W}^{n,m}_{\binom{n+d}{d}}$.*

*Proof.* By Theorem 5.6, it suffices to show that there is some $W \in \mathcal{W}^{n,m}_{\binom{n+d}{d}}$ such that the non-bias Vandermonde matrix of $W$ is non-singular; see Appendix A.2 for proof details. □

**Remark 5.8.** The proof of Corollary 5.7 still holds even if we restrict every non-bias weight $w^{(1)}_{i,j}$ in the first layer to satisfy $|w^{(1)}_{i,j}| < \lambda$ for some fixed constant $\lambda > 0$.

For the rest of this section, let $\{\lambda_k\}_{k \in \mathbb{N}}$ be a divergent increasing sequence of positive real numbers, and let $\{Y_k\}_{k \in \mathbb{N}}$ be a sequence of closed intervals of $\mathbb{R}$, such that $Y_{k'} \subseteq Y_k$ whenever $k' \leq k$, and such that each interval $Y_k = [y'_k, y''_k]$ has length $\lambda_k$. Let $d \geq 1$ be an integer, and suppose $\sigma \in \mathcal{C}(\mathbb{R})$. For each $k \in \mathbb{N}$, let $\sigma_k$ be the best polynomial approximant to $\sigma|_{Y_k}$ of degree $d$. Given $r > 0$ and any integer $N \geq 1$, define the closed ball $B^N_r := \{x \in \mathbb{R}^N : \|x\|_2 \leq r\}$.

**Lemma 5.9.** *If $d \geq 2$, $\lim_{k \to \infty} E_d(\sigma|_{Y_k}) = \infty$, and $\lambda_k \in \Omega(k^\gamma)$ for some $\gamma > 0$, then for every $\varepsilon > 0$, there is a subsequence $\{k_t\}_{t \in \mathbb{N}}$ of $\mathbb{N}$, and a sequence $\{y_{k_t}\}_{t \in \mathbb{N}}$ of real numbers, such that $y'_{k_t} < y_{k_t} < y''_{k_t}$, $\sigma(y_{k_t}) = \sigma_{k_t}(y_{k_t})$, and*

$$\frac{\min\{|y_{k_t} - y'_{k_t}|, |y_{k_t} - y''_{k_t}|\}}{|y'_{k_t} - y''_{k_t}|} > \frac{1}{d+1} - \varepsilon,$$

*for all $t \in \mathbb{N}$. (See Appendix B for proof details.)*

The proofs of the next three lemmas can be found in Appendix C.

**Lemma 5.10.** *For any constant $\gamma > 0$,*

$$\lim_{k \to \infty} \frac{\|\sigma_k - \sigma\|_{\infty, Y_k}}{(\lambda_k)^{1+\gamma}} = 0.$$

**Lemma 5.11.** *Let $K \geq N \geq 1$ be integers, let $r_0, \ldots, r_N \geq 1$ be fixed real numbers, and let $S(\lambda)$ be a set $\{p_0(\lambda), \ldots, p_N(\lambda)\}$ of $N+1$ affinely independent points in $\mathbb{R}^K$, parametrized by $\lambda > 0$, where each point $p_i(\lambda)$ has (Cartesian) coordinates $(\lambda^{r_i} p_{i,1}, \ldots, \lambda^{r_i} p_{i,K})$ for some fixed non-zero scalars $p_{i,1}, \ldots, p_{i,K}$. Let $\Delta(\lambda)$ be the convex hull of $S(\lambda)$, i.e. $\Delta(\lambda)$ is an $N$-simplex, and for each $0 \leq i \leq N$, let $h_i(\lambda)$ be the height of $\Delta(\lambda)$ w.r.t. apex $p_i(\lambda)$. Let $h(\lambda) := \max\{h_i(\lambda) : 0 \leq i \leq N\}$ and $r_{\min} := \min\{r_1, \ldots, r_N\}$. If $r_j > r_{\min}$ for some $0 \leq j \leq N$, then there exists some $\gamma > 0$ such that $h(\lambda) \in \Omega(\lambda^{r_{\min}+\gamma})$.*

**Lemma 5.12.** *Let $M, N \geq 1$ be integers, let $\tau > 0$, and let $0 < \theta < 1$. Suppose $\varphi : \mathbb{R}^M \to \mathbb{R}^N$ is a continuous open map such that $\varphi(\mathbf{0}_M) = \mathbf{0}_N$, and $\varphi(\lambda x) \geq \lambda \varphi(x)$ for all $x \in \mathbb{R}^M$, $\lambda > 0$. Let $\{U_k\}_{k \in \mathbb{N}}$ be a sequence where each $U_k$ is a dense subspace of $B^M_{\lambda_k} \backslash B^M_{\theta \lambda_k}$. Then for every $\delta > 0$, there exists some (sufficiently large) $k \in \mathbb{N}$, and some points $u_0, \ldots, u_N$ in $U_k$, such that for each point $p \in B^N_\tau$, there are scalars $b_0, \ldots, b_N \geq 0$ satisfying $p = \sum_{i=0}^N b_i \varphi(u_i)$, $b_0 + \cdots + b_N = 1$, and $|b_i - \frac{1}{N}| < \delta$ for all $0 \leq i \leq N$.*

**Outline of strategy for proving Theorem 3.1.** The first crucial insight is that $\mathcal{P}_{\leq d}(\mathbb{R}^n)$, as a real vector space, has dimension $\binom{n+d}{d}$. Our strategy is to consider $N = \binom{n+d}{d}$ hidden units. Every hidden unit represents a continuous function $g_j : X \to \mathbb{R}$ determined by its weights $W$ and the activation function $\sigma$. If $g_1, \ldots, g_N$ can be well-approximated (on $X$) by linearly independent polynomial functions in $\mathcal{P}_{\leq d}(\mathbb{R}^n)$, then we can choose suitable linear combinations of these $N$ functions to approximate all coordinate functions $f^{[t]}$ (independent of how large $m$ is). To approximate each $g_j$, we consider a suitable sequence $\{\sigma_{\lambda_k}\}_{k=1}^\infty$ of degree $d$ polynomial approximations to $\sigma$, so that $g_j$ is approximated by a sequence of degree $d$ polynomial functions $\{\widehat{g}^W_{j,k}\}_{k=1}^\infty$. We shall also vary $W$ concurrently with $k$, so that $\|\widehat{\mathbf{w}}^{(1)}_j\|_2$ increases together with $k$. By Corollary 5.7, the weights can always be chosen so that $\widehat{g}^W_{1,k}, \ldots, \widehat{g}^W_{N,k}$ are linearly independent.

The second crucial insight is that every function in $\mathcal{P}_{\leq d}(\mathbb{R}^n)$ can be identified geometrically as a point in Euclidean $\binom{n+d}{d}$-space. We shall choose the bias weights so that $\widehat{g}^W_{1,k}, \ldots, \widehat{g}^W_{N,k}$ correspond

to points on a hyperplane, and we shall consider the barycentric coordinates of the projections of both $f^{[t]}$ and the constant function onto this hyperplane, with respect to $\widehat{g}_{1,k}^W, \ldots, \widehat{g}_{N,k}^W$. As the values of $k$ and $\|\widehat{\mathbf{w}}_j^{(1)}\|_2$ increase, both projection points have barycentric coordinates that approach $(\frac{1}{N}, \ldots, \frac{1}{N})$, and their difference approaches $\mathbf{0}$; cf. Lemma 5.12. This last observation, in particular, when combined with Lemma 5.9 and Lemma 5.10, is a key reason why the minimum number $N$ of hidden units required for the UAP to hold is independent of the approximation error threshold $\varepsilon$.

**Proof of Theorem 3.1.** Fix some $\varepsilon > 0$, and for brevity, let $N = \binom{n+d}{d}$. Theorem 3.1 is trivially true when $f$ is constant, so assume $f$ is non-constant. Fix a point $x_0 \in X$, and define $f_{\mathbf{0}} \in \mathcal{C}(X, \mathbb{R}^m)$ by $f_{\mathbf{0}}^{[t]} := f^{[t]} - f^{[t]}(x_0)$ for all $1 \leq t \leq m$. Next, let $r_X(x_0) := \sup\{\|x - x_0\|_2 : x \in X\}$, and note that $r_X(x_0) < \infty$, since $X$ is compact. By replacing $X$ with a closed tubular neighborhood of $X$ if necessary, we may assume without loss of generality that $r_X(x_0) > 0$.

Define $\{\lambda_k\}_{k \in \mathbb{N}}$, $\{Y_k\}_{k \in \mathbb{N}}$ and $\{\sigma_k\}_{k \in \mathbb{N}}$ as before, with an additional condition that $\lambda_k \in \Omega(k^\tau)$ for some $\tau > 0$. Assume without loss of generality that there exists a sequence $\{y_k\}_{k \in \mathbb{N}}$ of real numbers, such that $y_k' < y_k < y_k''$, $\sigma(y_k) = \sigma_k(y_k)$, and

$$\frac{\min\{|y_k - y_k'|, |y_k - y_k''|\}}{\lambda_k} = \frac{\min\{|y_k - y_k'|, |y_k - y_k''|\}}{|y_k' - y_k''|} > \frac{1}{d+2}, \tag{3}$$

for all $k \in \mathbb{N}$. The validity of this assumption in the case $\lim_{k \to \infty} E_d(\sigma|_{Y_k}) = \infty$ is given by Lemma 5.9. If instead $\lim_{k \to \infty} E_d(\sigma|_{Y_k}) < \infty$, then as $k \to \infty$, the sequence $\{\sigma_k\}_{k \in \mathbb{N}}$ converges to some $\widehat{\sigma} \in \mathcal{P}_{\leq d}(\mathbb{R})$. Hence, the assumption is also valid in this case, since for any $\widehat{y} \in \mathbb{R}$ such that $\sigma(\widehat{y}) = \widehat{\sigma}(\widehat{y})$, we can always choose $\{Y_k\}_{k \in \mathbb{N}}$ to satisfy $\frac{y_k' + y_k''}{2} = \widehat{y}$ for all $k \in \mathbb{N}$, which then allows us to choose $\{y_k\}_{k \in \mathbb{N}}$ that satisfies $\lim_{k \to \infty} \frac{\min\{|y_k - y_k'|, |y_k - y_k''|\}}{\lambda_k} = \frac{1}{2} > \frac{1}{d+2}$.

By Lemma 5.10, we may further assume that $\|\sigma_k - \sigma\|_{\infty, Y_k} < \frac{\varepsilon(\lambda_k)^{1+\gamma}}{C}$ for all $k \in \mathbb{N}$, where $C > 0$ and $\gamma > 0$ are constants whose precise definitions we give later. Also, for any $W \in \mathcal{W}_N^{n,m}$, we can choose $\sigma' \in \mathcal{C}(\mathbb{R})$ that is arbitrarily close to $\sigma$ in the uniform metric, such that $\|\rho_W^\sigma - \rho_W^{\sigma'}\|_{\infty, X}$ is arbitrarily small. Since $\sigma \in \mathcal{C}(\mathbb{R}) \backslash \mathcal{P}_{\leq d-1}(\mathbb{R})$ by assumption, we may hence perturb $\sigma$ if necessary, and assume without loss of generality that every $\sigma_k$ is a polynomial of degree $d$ with all-non-zero coefficients, such that $\sigma_k(y_k) \neq 0$.

For every $r > 0$ and $k \in \mathbb{N}$, let $\mathcal{W}_r' := \{W \in \mathcal{W}_N^{n,m} : \|\widehat{\mathbf{w}}_j^{(1)}\|_2 \leq r \text{ for all } 1 \leq j \leq N\}$, and define

$$\lambda_k' := \sup\left\{r > 0 : \{y_k + \widehat{\mathbf{w}}_j^{(1)} \cdot (x - x_0) \in \mathbb{R} : x \in X, W \in \mathcal{W}_r'\} \subseteq Y_k \text{ for all } 1 \leq j \leq N\right\}.$$

Each $\lambda_k'$ is well-defined, since $r_X(x_0) < \infty$. Note also that $\lambda_k' r_X(x_0) = \min\{|y_k - y_k'|, |y_k - y_k''|\}$ by definition, hence it follows from (3) that $\frac{\lambda_k}{\lambda_k'} < (d+2) r_X(x_0)$. In particular, $\{\lambda_k'\}_{k \in \mathbb{N}}$ is a divergent increasing sequence of positive real numbers.

Given any $p \in \mathcal{P}_{\leq d}(\mathbb{R}^n)$, let $\nu(p) \in \mathbb{R}^N$ denote the vector of coefficients with respect to the basis $\{q_1(x - x_0), \ldots, q_N(x - x_0)\}$ (i.e. if $\nu(p) = (\nu_1, \ldots, \nu_N)$, then $p(x) = \sum_{1 \leq i \leq N} \nu_i q_i(x - x_0)$), and let $\widehat{\nu}(p) \in \mathbb{R}^{N-1}$ be the truncation of $\nu(p)$ by removing the first coordinate. Note that $q_1(x)$ is the constant monomial, so this first coordinate $\nu_1$ is the coefficient of the constant term. For convenience, let $\nu_i(p)$ (resp. $\widehat{\nu}_i(p)$) be the $i$-th entry of $\nu(p)$ (resp. $\widehat{\nu}(p)$).

For each $k \in \mathbb{N}$, $W \in \mathcal{W}_{\lambda_k'}'$, $1 \leq j \leq N$, define functions $g_{j,k}^W, \widehat{g}_{j,k}^W$ in $\mathcal{C}(X)$ by $x \mapsto \sigma(\mathbf{w}_j^{(1)} \cdot (1, x))$ and $x \mapsto \sigma_k(\mathbf{w}_j^{(1)} \cdot (1, x))$ respectively. By definition, $\nu_i(\widehat{g}_{j,k}^W)$ can be treated as a function of $W$, and note that $\nu_i(\widehat{g}_{j,k}^{\lambda W}) = \lambda^{\deg q_i} \nu_i(\widehat{g}_{j,k}^W)$ for any $\lambda > 0$. (Here, $\deg q_i$ denotes the total degree of $q_i$.) Since $\deg q_i = 0$ only if $i = 1$, it then follows that $\widehat{\nu}_i(\widehat{g}_{j,k}^{\lambda W}) \geq \lambda \widehat{\nu}_i(\widehat{g}_{j,k}^W)$ for all $\lambda > 0$.

For each $k \in \mathbb{N}$, define the "shifted" function $\sigma_k' : Y_k \to \mathbb{R}$ by $y \mapsto \sigma_k(y + y_k)$. Next, let $\mathcal{W}_k'' := {}^{\sigma_k'} \mathcal{W}_{n,N;x_0}^{\text{ind}} \cap (\mathcal{W}_{\lambda_k'}' \backslash \mathcal{W}_{0.5\lambda_k'}')$, and suppose $W \in \mathcal{W}_k''$. Note that in the definition of $\mathcal{W}_k''$, we do not impose any restrictions on the bias weights. Thus, given any such $W$, we could choose the bias weights of $W^{(1)}$ to be $w_{j,0}^{(1)} = y_k - \widehat{\mathbf{w}}_j^{(1)} \cdot x_0$ for all $1 \leq j \leq N$. This implies that each $\widehat{g}_{j,k}^W$ represents the map $x \mapsto \sigma_k(\widehat{\mathbf{w}}_j^{(1)} \cdot (x - x_0) + y_k)$, hence $\widehat{g}_{j,k}^W(x_0) = \sigma_k(y_k) = \sigma(y_k)$. Consequently,

by the definitions of $Y_k$ and $\mathcal{W}'_{\lambda'_k}$, we infer that

$$\|g^W_{j,k} - \widehat{g}^W_{j,k}\|_{\infty,X} < \frac{\varepsilon(\lambda_k)^{1+\gamma}}{C}. \tag{4}$$

By Corollary 5.7 and Remark 5.3, $\mathcal{W}''_k$ is dense in $(\mathcal{W}'_{\lambda'_k} \setminus \mathcal{W}'_{0.5\lambda'_k})$, so such a $W$ exists (with its bias weights given as above). By the definition of $\sigma'_k \mathcal{W}^{\text{ind}}_{n,N;x_0}$, we infer that $\{\widehat{g}^W_{1,k}, \ldots, \widehat{g}^W_{N,k}\}$ is linearly independent and hence spans $\mathcal{P}_{\leq d}(X)$. Thus, for every $1 \leq t \leq m$, there exist $a^{[t]}_{1,k}, \ldots, a^{[t]}_{N,k} \in \mathbb{R}$, which are uniquely determined once $k$ is fixed, such that $f^{[t]}_{\mathbf{0}} = a^{[t]}_{1,k}\widehat{g}^W_{1,k} + \cdots + a^{[t]}_{N,k}\widehat{g}^W_{N,k}$. Evaluating both sides of this equation at $x = x_0$, we then get

$$a^{[t]}_{1,k} + \cdots + a^{[t]}_{N,k} = 0. \tag{5}$$

For each $\ell \in \mathbb{R}$, define the hyperplane $\mathcal{H}_\ell := \{(u_1, \ldots, u_N) \in \mathbb{R}^N : u_1 = \ell\}$. Recall that $q_1(x)$ is the constant monomial, so the first coordinate of each $\nu(\widehat{g}^W_{j,k})$ equals $\sigma(y_k)$, which implies that $\nu(\widehat{g}^W_{1,k}), \ldots, \nu(\widehat{g}^W_{N,k})$ are $N$ points on $\mathcal{H}_{\sigma(y_k)} \cong \mathbb{R}^{N-1}$. Let $c_f := \max\{\|\widehat{\nu}(f^{[t]})\|_2 : 1 \leq t \leq m\}$. (This is non-zero, since $f$ is non-constant.) Note that $\mathbf{0}_{N-1}$ and $\widehat{\nu}(f^{[t]})$ (for all $t$) are points in $B^{N-1}_{c_f}$. So for any $\delta > 0$, Lemma 5.12 implies that there exists some sufficiently large $k \in \mathbb{N}$ such that we can choose some $W \in \mathcal{W}''_k$, so that there are non-negative scalars $b^{[t]}_{j,k}, b'_{j,k}$ (for $1 \leq j \leq N$, $1 \leq t \leq m$) contained in the interval $(\frac{1}{N} - \delta, \frac{1}{N} + \delta)$ that satisfy the following:

$$b^{[t]}_{1,k} + \cdots + b^{[t]}_{N,k} = b'_{1,k} + \cdots + b'_{N,k} = 1 \quad \text{(for all } 1 \leq t \leq m\text{)};$$

$$\mathbf{0}_{N-1} = \sum_{j=1}^N b'_{j,k}\widehat{\nu}(\widehat{g}^W_{j,k}); \quad \widehat{\nu}(f^{[t]}) = \sum_{j=1}^N b^{[t]}_{j,k}\widehat{\nu}(\widehat{g}^W_{j,k}) \quad \text{(for all } 1 \leq t \leq m\text{)}.$$

Note that $\nu(f^{[t]}_{\mathbf{0}} + \sigma(y_k)) = b^{[t]}_{1,k}\nu(\widehat{g}^W_{1,k}) + \cdots + b^{[t]}_{N,k}\nu(\widehat{g}^W_{N,k})$ and $(\mathbf{0}_{N-1}, \sigma(y_k)) = b'_{1,k}\nu(\widehat{g}^W_{1,k}) + \cdots + b'_{N,k}\nu(\widehat{g}^W_{N,k})$, so we get

$$f^{[t]}_{\mathbf{0}} = (b^{[t]}_{1,k} - b'_{1,k})\widehat{g}^W_{1,k} + \cdots + (b^{[t]}_{N,k} - b'_{N,k})\widehat{g}^W_{N,k}.$$

Since $a^{[t]}_{1,k}, \ldots, a^{[t]}_{N,k}$ are unique (for fixed $k$), we infer that $a^{[t]}_{j,k} = b^{[t]}_{j,k} - b'_{j,k}$ for each $1 \leq j \leq N$. Thus, for this sufficiently large $k$, it follows from $b^{[t]}_{j,k}, b'_{j,k} \in (\frac{1}{N} - \delta, \frac{1}{N} + \delta)$ that

$$a^{[t]}_{j,k} \geq (\tfrac{1}{N} - \delta) - (\tfrac{1}{N} + \delta) \geq -2\delta. \tag{6}$$

Let $S_k := \{\widehat{\nu}(\widehat{g}^W_{1,k}), \ldots, \widehat{\nu}(\widehat{g}^W_{N,k})\}$, let $\Delta_k$ be the convex hull of $S_k$, and for each $j$, let $h_j(\Delta_k)$ be the height of $\Delta_k$ w.r.t. apex $\widehat{\nu}(\widehat{g}^W_{j,k})$. Let $h(\Delta_k) := \max\{h_j(\Delta_k) : 1 \leq j \leq N\}$. Since $\widehat{\nu}_i(\widehat{g}^{\lambda W}_{j,k}) = \lambda^{\deg q_i}\widehat{\nu}_i(\widehat{g}^W_{j,k})$ for all $i$, and since $d \geq 2$ (i.e. $\deg q_N > 1$), it follows from Lemma 5.11 that there exists some $\gamma > 0$ such that $h(\Delta_k) \in \Omega((\lambda'_k)^{1+\gamma})$. Using this particular $\gamma > 0$, we infer that there exists some constant $0 < C' < \infty$ such that $\frac{(\lambda'_k)^{1+\gamma}}{h(\Delta_k)} < C'$ for all sufficiently large $k$.

Note that $2\delta$ is an upper bound of the normalized difference for each barycentric coordinate of the two points $\widehat{\nu}(f^{[t]})$ and $\mathbf{0}_{N-1}$ (contained in $B^{N-1}_{c_f}$), which satisfies

$$2\delta \leq \frac{c_f}{h(\Delta_k)} = \frac{c_f}{(\lambda_k)^{1+\gamma}} \cdot \left(\frac{\lambda_k}{\lambda'_k}\right)^{1+\gamma} \cdot \frac{(\lambda'_k)^{1+\gamma}}{h(\Delta_k)} < \frac{c_f}{(\lambda_k)^{1+\gamma}}[(d+2)r_X(x_0)]^{1+\gamma}C'. \tag{7}$$

Now, define $C := 2Nc_f[(d+2)r_X(x_0)]^{1+\gamma}C' > 0$. Thus, for sufficiently large $k$, it follows from (5), (6) and (7) that

$$|a^{[t]}_{1,k}| + \cdots + |a^{[t]}_{N,k}| \leq a^{[t]}_{1,k} + \cdots + a^{[t]}_{N,k} + 4N\delta = 4N\delta \leq \frac{C}{(\lambda_k)^{1+\gamma}} \tag{8}$$

For this sufficiently large $k$, define $g \in \mathcal{C}(X, \mathbb{R}^m)$ by $g^{[t]} = a^{[t]}_{1,k}g^W_{1,k} + \cdots + a^{[t]}_{N,k}g^W_{N,k}$ for each $t$. Using (4) and (8), it follows that

$$\|f^{[t]}_{\mathbf{0}} - g^{[t]}\|_{\infty,X} = \|a^{[t]}_{1,k}(g^W_{1,k} - \widehat{g}^W_{1,k}) + \cdots + a^{[t]}_{N,k}(g^W_{N,k} - \widehat{g}^W_{N,k})\|_{\infty,X}$$

$$\leq |a^{[t]}_{1,k}| \cdot \|g^W_{1,k} - \widehat{g}^W_{1,k}\|_{\infty,X} + \cdots + |a^{[t]}_{N,k}| \cdot \|g^W_{N,k} - \widehat{g}^W_{N,k}\|_{\infty,X}$$

$$< \varepsilon.$$

Finally, for all $1 \leq t \leq m$, let $w_{j,t}^{(2)} = a_{j,k}^{[t]}$ for each $1 \leq j \leq N$, and let $w_{0,t}^{(2)} = f^{[t]}(x_0)$. This gives $\rho_W^{\sigma[t]} = g^{[t]} + f^{[t]}(x_0)$. Therefore, the identity $f^{[t]} = f_{\mathbf{0}}^{[t]} + f^{[t]}(x_0)$ implies $\|f - \rho_W^\sigma\|_{\infty,X} < \varepsilon$.

Notice that for all $\delta > 0$, we showed in (6) that there is a sufficiently large $k$ such that $a_{j,k}^{[t]} \geq -2\delta$. A symmetric argument yields $a_{j,k}^{[t]} \leq 2\delta$. Thus, for all $\lambda > 0$, we can choose $W$ so that all non-bias weights in $W^{(2)}$ are contained in the interval $(-\lambda, \lambda)$; this proves assertion (i) of the theorem.

Note also that we do not actually require $\delta > 0$ to be arbitrarily small. Suppose instead that we choose $k \in \mathbb{N}$ sufficiently large, so that the convex hull of $S_k$ contains $\mathbf{0}_{N-1}$ and $\widehat{\nu}(f^{[t]})$ (for all $t$). In this case, observe that our choice of $k$ depends only on $f$ (via $\widehat{\nu}(f^{[t]})$) and $\sigma$ (via the definition of $\{\lambda_k\}_{k \in \mathbb{N}}$). The inequality (7) still holds for any $\delta$ satisfying $b_{j,k}^{[t]}, b'_{j,k} \in (\frac{1}{N} - \delta, \frac{1}{N} + \delta)$ for all $j, t$. Thus, our argument to show $\|f - \rho_W^\sigma\|_{\infty,X} < \varepsilon$ holds verbatim, which proves assertion (ii). $\qquad \square$

**Proof of Theorem 3.2.** Fix some $\varepsilon > 0$, and consider an arbitrary $t \in \{1, \ldots, m\}$. For each integer $d \geq 1$, let $p_d^{[t]}$ be the best polynomial approximant to $f^{[t]}$ of degree $d$. By Theorem 2.2, we have $\|f^{[t]} - p_d^{[t]}\|_{\infty,X} \leq 6\omega_{f^{[t]}}(\frac{D}{2d})$ for all $d \geq 1$, hence it follows from the definition of $d_\varepsilon$ that

$$\|f^{[t]} - p_{d_\varepsilon}^{[t]}\|_{\infty,X} \leq 6\omega_{f^{[t]}}\left(\frac{D}{2d_\varepsilon}\right) < \varepsilon.$$

Define $\varepsilon' := \varepsilon - \max\{6\omega_{f^{[t]}}(\frac{D}{2d_\varepsilon}) : 1 \leq t \leq m\}$. Note that $\varepsilon' > 0$, and $\|f^{[t]} - p_{d_\varepsilon}^{[t]}\|_{\infty,X} \leq \varepsilon - \varepsilon'$ (for all $1 \leq t \leq m$). By Theorem 3.1, there exists some $W \in \mathcal{W}_{\binom{n+d_\varepsilon}{d_\varepsilon}}$ satisfying $\|p_{d_\varepsilon}^{[t]} - \rho_W^{\sigma[t]}\|_{\infty,X} < \varepsilon'$ for all $1 \leq t \leq m$, which implies

$$\|f^{[t]} - \rho_W^{\sigma[t]}\|_{\infty,X} \leq \|f^{[t]} - p_{d_\varepsilon}^{[t]}\|_{\infty,X} + \|p_{d_\varepsilon}^{[t]} - \rho_W^{\sigma[t]}\|_{\infty,X} < (\varepsilon - \varepsilon') + \varepsilon' = \varepsilon,$$

therefore $\|f - \rho_W^\sigma\|_{\infty,X} < \varepsilon$. Conditions (i) and (ii) follow from Theorem 3.1. Finally, note that $\omega_{f^{[t]}}(\frac{D}{2d}) \in \mathcal{O}(\frac{1}{d})$ (for fixed $D$), i.e. $d_\varepsilon \in \mathcal{O}(\frac{1}{\varepsilon})$, hence $\binom{n+d_\varepsilon}{d_\varepsilon} = \frac{n(n-1)\ldots(n-d_\varepsilon+1)}{n!} \in \mathcal{O}(\varepsilon^{-n})$. $\quad \square$

**Proof of Theorem 3.3.** Most of the work has already been done earlier in the proofs of Theorem 3.1 and Theorem 3.2. The key observation is that $\det(Q[W])$ is a non-zero polynomial in terms of the weights $W$, hence $\{\det(Q[W]) \neq 0 : W \in \mathcal{W}_{\binom{n+d}{d}}\}$ is dense in $\mathcal{W}_{\binom{n+d}{d}}$, or equivalently, its complement has Lebesgue measure zero. $\qquad \square$

## 6 Conclusion and Further Remarks

Theorem 5.6 is rather general, and could potentially be used to prove analogs of the universal approximation theorem for other classes of neural networks, such as convolutional neural networks and recurrent neural networks. In particular, finding a *single* suitable set of weights (as a representative of the infinitely many possible sets of weights in the given class of neural networks), with the property that its corresponding "non-bias Vandermonde matrix" (see Definition 5.5) is non-singular, would serve as a straightforward criterion for showing that the UAP holds for the given class of neural networks (with certain weight constraints). We formulated this criterion to be as general as we could, with the hope that it would applicable to future classes of "neural-like" networks.

We believe our algebraic approach could be emulated to eventually yield a unified understanding of how depth, width, constraints on weights, and other architectural choices, would influence the approximation capabilities of arbitrary neural networks.

Finally, we end our paper with an open-ended question. The proofs of our results in Section 5 seem to suggest that non-bias weights and bias weights play very different roles. We could impose very strong restrictions on the non-bias weights and still have the UAP. What about the bias weights?

ACKNOWLEDGMENTS

This research is supported by the National Research Foundation, Singapore, under its NRFF program (NRFFAI1-2019-0005).

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

## A  GENERALIZED WRONSKIANS AND THE PROOF OF THEOREM 5.6

First, we recall the notion of generalized Wronskians as given in (LeVeque, 1956, Chap. 4.3). Let $\Delta_0, \ldots, \Delta_{N-1}$ be any $N$ differential operators of the form

$$\Delta_k = \left(\frac{\partial}{\partial x_1}\right)^{\alpha_1} \cdots \left(\frac{\partial}{\partial x_n}\right)^{\alpha_n}, \text{ where } \alpha_1 + \cdots + \alpha_n \leq k.$$

Let $f_1, \ldots, f_N \in \mathcal{P}(\mathbb{R}^n)$. The *generalized Wronskian* of $(f_1, \ldots, f_N)$ associated to $\Delta_0, \ldots, \Delta_{N-1}$ is defined as the determinant of the matrix $M = [\Delta_{i-1} f_j(x)]_{1 \leq i,j \leq N}$. In general, $(f_1, \ldots, f_N)$ has multiple generalized Wronskians, corresponding to multiple choices for $\Delta_0, \ldots, \Delta_{N-1}$.

### A.1  PROOF OF THEOREM 5.6

For brevity, let $N = \binom{n+d}{d}$ and let $\mathbf{x} = (x_1, \ldots, x_n)$. Recall that $\lambda_1 < \cdots < \lambda_N$ are all the $n$-tuples in $\Lambda_{\leq d}^n$ in the colexicographic order. For each $1 \leq i, k \leq N$, write $\lambda_k = (\lambda_{k,1}, \ldots, \lambda_{k,n})$, define the differential operator $\Delta_{\lambda_k} = \left(\frac{\partial}{\partial x_1}\right)^{\lambda_{k,1}} \cdots \left(\frac{\partial}{\partial x_n}\right)^{\lambda_{k,n}}$, and let $\alpha_{\lambda_k}^{(i)}$ be the coefficient of the monomial $q_k(\mathbf{x})$ in $\Delta_{\lambda_i} p(\mathbf{x})$. Consider an arbitrary $W \in \mathcal{U}$, and for each $1 \leq j \leq N$, define $f_j \in \mathcal{P}_{\leq d}(\mathbb{R}^n)$ by the map $\mathbf{x} \mapsto p(w_{1,j}^{(1)} x_1, \ldots, w_{n,j}^{(1)} x_n)$. Note that $\mathcal{F}_{p,\mathbf{0}_n}(W) = (f_1, \ldots, f_N)$ by definition. Next, define the matrix $M_W(\mathbf{x}) := [\Delta_i f_j(x)]_{1 \leq i,j \leq N}$, and note that $\det M_W(\mathbf{x})$ is the generalized Wronskian of $(f_1, \ldots, f_N)$ associated to $\Delta_1, \ldots, \Delta_N$. In particular, this generalized Wronskian is well-defined, since the definition of the colexicographic order implies that $\lambda_{k,1} + \cdots + \lambda_{k,n} \leq k$ for all possible $k$. Similar to the univariate case, $(f_1, \ldots, f_N)$ is linearly independent if (and only if) its generalized Wronskian is not the zero function (Wolsson, 1989). Thus, to show that $W \in {}^P\mathcal{U}^{\text{ind}}$, it suffices to show that the evaluation $\det M_W(\mathbf{1}_n)$ of this generalized Wronskian at $\mathbf{x} = \mathbf{1}_n$ gives a non-zero value, where $\mathbf{1}_n$ denotes the all-ones vector in $\mathbb{R}^n$.

Observe that the $(i,j)$-th entry of $M_W(\mathbf{1}_n)$ equals $(\widehat{\mathbf{w}}_j^{(1)})^{\lambda_i}(\Delta_{\lambda_i} p)(\widehat{\mathbf{w}}_j^{(1)})$, hence we can check that $M_W(\mathbf{1}_n) = M'M''$, where $M'$ is an $N$-by-$N$ matrix whose $(i,j)$-th entry is given by

$$M'_{i,j} = \begin{cases} \alpha_{\lambda_j - \lambda_i}^{(i)}, & \text{if } \lambda_j - \lambda_i \in \Lambda_{\leq d}^n; \\ 0, & \text{if } \lambda_j - \lambda_i \notin \Lambda_{\leq d}^n; \end{cases}$$

and where $M'' = Q[W]$ is the non-bias Vandermonde matrix of $W$.

It follows from the definition of the colexicographic order that $\lambda_j - \lambda_i$ necessarily contains at least one strictly negative entry whenever $j < i$, hence we infer that $M'$ is upper triangular. The diagonal entries of $M'$ are $\alpha_{\mathbf{0}_n}^{(1)}, \alpha_{\mathbf{0}_n}^{(2)}, \ldots, \alpha_{\mathbf{0}_n}^{(N)}$, and note that $\alpha_{\mathbf{0}_n}^{(i)} = (\lambda_{i,1}! \cdots \lambda_{i,n}!) \alpha_{\lambda_i}^{(1)}$ for each $1 \leq i \leq N$, where $\lambda_{i,1}! \cdots \lambda_{i,n}!$ denotes the product of the factorials of the entries of the $n$-tuple $\lambda_i$. In particular, $\lambda_{i,1}! \cdots \lambda_{i,n}! \neq 0$, and $\alpha_{\lambda_i}^{(1)}$, which is the coefficient of the monomial $q_i(\mathbf{x})$ in $p(\mathbf{x})$, is non-zero. Thus, $\det(M') \neq 0$.

We have come to the crucial step of our proof. If we can show that $\det(M'') = \det(Q[W]) \neq 0$, then $\det(M_W(\mathbf{1}_n)) = \det(M') \det(M'') \neq 0$, and hence we can infer that $W \in {}^P\mathcal{U}^{\text{ind}}$. This means that ${}^P\mathcal{U}^{\text{ind}}$ contains the subset $\mathcal{U}' \subseteq \mathcal{U}$ consisting of all $W$ such that $Q[W]$ is non-singular. Note that $\det(Q[W])$ is a polynomial in terms of the non-bias weights in $W^{(1)}$ as its variables, so we could write this polynomial as $r = r(W)$. Consequently, if we can find a single $W \in \mathcal{U}$ such that $Q[W]$ is non-singular, then $r(W)$ is not identically zero on $\mathcal{U}$, which then implies that $\mathcal{U}' = \{W \in \mathcal{U} : r(W) \neq 0\}$ is dense in $\mathcal{U}$ (w.r.t. the Euclidean metric). $\qquad\square$

## A.2 Proof of Corollary 5.7

Let $N := \binom{n+d}{d}$. By Theorem 5.6, it suffices to show that there exists some $W \in \mathcal{W}_N^{n,m}$ such that the non-bias Vandermonde matrix of $W$ is non-singular. Consider $W \in \mathcal{W}_N^{n,m}$ such that $w_{i,j}^{(1)} = (w_{1,j}^{(1)})^{(d+1)^i}$. Recall that the monomials in $\mathcal{M}_{\leq d}^n$ are arranged in colexicographic order, i.e.

$$1, x_1, x_1^2, \ldots, x_1^d, x_2, x_1 x_2, x_1^2 x_2, \ldots, x_2^2, x_1 x_2^2, \ldots, x_n^d.$$

Thus, there are fixed integers $0 = \beta_1 < \beta_2 < \cdots < \beta_N$, such that the $(i, j)$-th entry of $Q[W]$ is $(w_{1,j}^{(1)})^{\beta_i}$. Such matrices are well-studied in algebraic combinatorics, and the determinant of $Q[W]$ is a *Schur polynomial*; see (Stanley, 1999). In particular, if we choose positive pairwise distinct values for $w_{1,j}^{(1)}$ (for $1 \leq j \leq N$), then $Q[W]$ is non-singular, since a Schur polynomial can be expressed as a (non-negative) sum of certain monomials; see (Stanley, 1999, Sec. 7.10) for details. □

## B An analog of Kadec's theorem and the proof of Lemma 5.9

Throughout this section, suppose $\sigma \in \mathcal{C}(\mathbb{R})$ and let $d \geq 1$ be an integer. We shall use the same definitions for $\{\lambda_k\}_{k\in\mathbb{N}}$, $\{Y_k\}_{k\in\mathbb{N}}$ and $\{\sigma_k\}_{k\in\mathbb{N}}$ as given immediately after Remark 5.8. Our goal for this section is to prove Theorem B.1 below, so that we can infer Lemma 5.9 as a consequence of Theorem B.1. Note that Theorem B.1 is an analog of the well-known Kadec's theorem (Kadec, 1960) from approximation theory. To prove Theorem B.1, we shall essentially follow the proof of Kadec's theorem as given in (Kadec, 1963).

We begin with a crucial observation. For every best polynomial approximant $\sigma_k$ to $\sigma|_{Y_k}$ of degree $d$, it is known that there are (at least) $d + 2$ values

$$y_k' \leq a_0^{(k)} < a_1^{(k)} < \cdots < a_{d+1}^{(k)} \leq y_k'',$$

and some sign $\delta_k \in \{\pm 1\}$, such that $\sigma(a_i^{(k)}) - \sigma_k(a_i^{(k)}) = (-1)^i \delta_k E_d(\sigma|_{Y_k})$ for all $0 \leq i \leq d+1$; see (Rivlin, 1981, Thm. 1.7). Define

$$\Delta_k := \max\left\{\left|\frac{a_i^{(k)} - y_k'}{y_k'' - y_k'} - \frac{i}{d+1}\right| : 0 \leq i \leq d+1\right\}.$$

**Theorem B.1.** *If $\lim_{k\to\infty} E_d(\sigma|_{Y_k}) = \infty$, then for any $\gamma > 0$, we have $\liminf_{k\to\infty} \frac{\Delta_k \lambda_k}{k^\gamma} = 0$.*

*Proof.* For every $k \in \mathbb{N}$, define the functions $e_k := \sigma - \sigma_k$ and $\phi_{k+1} := \sigma_k - \sigma_{k+1} = e_{k+1} - e_k$. Note that $e_k \in \mathcal{C}(\mathbb{R})$ and $\phi_{k+1} \in \mathcal{P}_{\leq d}(\mathbb{R})$. Since $y_{k+1}' \leq a_i^{(k)} \leq y_{k+1}''$ by assumption, it follows from the definition of $\sigma_{k+1}$ that $-E_d(\sigma|_{Y_{k+1}}) \leq e_{k+1}(a_i^{(k)}) \leq E_d(\sigma|_{Y_{k+1}})$. By the definition of $a_i^{(k)}$, we have $e_k(a_i^{(k)}) = (-1)^i \delta_k E_d(\sigma|_{Y_k})$. Consequently,

$$E_d(\sigma|_{Y_k}) - E_d(\sigma|_{Y_{k+1}}) \leq (-1)^i \delta_k (e_k - e_{k+1})(a_i^{(k)}) \leq E_d(\sigma|_{Y_k}) + E_d(\sigma|_{Y_{k+1}}),$$

or equivalently, $-E_d(\sigma|_{Y_k}) - E_d(\sigma|_{Y_{k+1}}) \leq (-1)^i \delta_k \phi_{k+1}(a_i^{(k)}) \leq E_d(\sigma|_{Y_{k+1}}) - E_d(\sigma|_{Y_k})$.

Since $Y_k \subseteq Y_{k+1}$ implies $E_d(\sigma|_{Y_k}) \leq E_d(\sigma|_{Y_{k+1}})$, it follows that $a_{2i-1} \leq a_i^{(k)} \leq a_{2i}$ (for each $0 \leq i \leq d+1$), where $a_{2i-1}$ and $a_{2i}$ are the roots of the equation $|\phi_{k+1}(y)| = E_d(\sigma|_{Y_{k+1}}) - E_d(\sigma|_{Y_k})$.

If $E_d(\sigma|_{Y_{k+1}}) = E_d(\sigma|_{Y_k})$, then $\sigma_{k+1} = \sigma_k$ by definition, so we could set $a_i^{(k+1)} = a_i^{(k)}$ for all $i$, i.e. there is nothing to prove in this case. Henceforth, assume $E_d(\sigma|_{Y_{k+1}}) \neq E_d(\sigma|_{Y_k})$, and consider the polynomial function

$$\phi(y) := \frac{\phi_{k+1}(y - y_k')}{E_d(\sigma|_{Y_{k+1}}) - E_d(\sigma|_{Y_k})}.$$

It then follows from (Kadec, 1963, Lem. 2) that

$$\Delta_k \leq \frac{\theta}{d+1} + \frac{1}{\lambda_k \sqrt{(d+1)\theta}} \operatorname{arcosh} \frac{E_d(\sigma|_{Y_{k+1}}) + E_d(\sigma|_{Y_k})}{E_d(\sigma|_{Y_{k+1}}) - E_d(\sigma|_{Y_k})}, \tag{9}$$

where $\theta$ is an arbitrary real number satisfying $0 < \theta < \frac{1}{2}$.

Since $\lim_{k\to\infty} E_d(\sigma|_{Y_k}) = \infty$ by assumption, the infinite product $\prod_{k=0}^{\infty} \dfrac{E_d(\sigma|_{Y_{k+1}})}{E_d(\sigma|_{Y_k})}$ diverges, and thus

the series $\sum_{k=0}^{\infty} \dfrac{E_d(\sigma|_{Y_{k+1}}) - E_d(\sigma|_{Y_k})}{E_d(\sigma|_{Y_{k+1}}) + E_d(\sigma|_{Y_k})}$ also diverges. It then follows from (9) that

$$\sum_{k=0}^{\infty} \frac{1}{\cosh\left[(\Delta_k - \frac{\theta}{d+1})\lambda_k \sqrt{(d+1)\theta}\right]} = \infty,$$

hence $\sum_{k=0}^{\infty} \dfrac{1}{(\Delta_k \lambda_k)^D} = \infty$ for any $D > 1$. If we compare the divergent series $\sum_{k=0}^{\infty} \dfrac{1}{(\Delta_k \lambda_k)^D}$ with

the convergent series $\sum_{k=0}^{\infty} \dfrac{1}{k^{1+\tau}}$ (for any $\tau > 0$), we thus get

$$\liminf_{k\to\infty} \frac{\Delta_k \lambda_k}{k^{(1+\tau)/D}} = 0.$$

Therefore, the assertion follows by letting $\gamma = \frac{1+\tau}{D}$. □

**Proof of Lemma 5.9.** Fix $\varepsilon > 0$. By Theorem B.1, we have $\liminf_{k\to\infty} \dfrac{\Delta_k \lambda_k}{k^\gamma} = 0$ for any $\gamma > 0$. Thus, by the definition of $\liminf$, there exists a subsequence $\{k'_t\}_{t\in\mathbb{N}}$ of $\mathbb{N}$ such that

$$\left|\frac{\Delta_{k'_t} \lambda_{k'_t}}{(k'_t)^\gamma}\right| < \varepsilon$$

for all $t \in \mathbb{N}$ (given any $\gamma > 0$). Since $\lambda_k$ is at least $\Omega(k^\gamma)$ for some $\gamma > 0$, we can use this particular $\gamma$ to get that $\liminf_{t\to\infty} \dfrac{\lambda_{k'_t}}{(k'_t)^\gamma} > 0$. Consequently, there is a subsequence $\{k_t\}_{t\in\mathbb{N}}$ of $\{k'_t\}_{t\in\mathbb{N}}$ such that $|\Delta_{k_t}| < \varepsilon$ for all $t \in \mathbb{N}$. Since $d \geq 2$ by assumption, it then follows that

$$\frac{1}{d+1} - \varepsilon < \frac{a_1^{(k_t)} - y'_{k_t}}{\lambda_{k_t}} < \frac{a_2^{(k_t)} - y'_{k_t}}{\lambda_{k_t}} < \frac{d}{d+1} + \varepsilon. \tag{10}$$

Now $\sigma - \sigma_{k_t}$ is continuous, so by the definition of $a_i^{(k_t)}$, there is some $a_1^{(k_t)} < y_{k_t} < a_2^{(k_t)}$ such that $\sigma(y_{k_t}) = \sigma_{k_t}(y_{k_t})$. From (10), we thus infer that $\dfrac{\min\{|y_{k_t} - y'_{k_t}|, |y_{k_t} - y''_{k_t}|\}}{\lambda_{k_t}} > \frac{1}{d+1} - \varepsilon$ as desired. □

## C PROOFS OF REMAINING LEMMAS

### C.1 PROOF OF LEMMA 5.10

Theorem 2.2 gives $\|\sigma_k - \sigma\|_{\infty, Y_k} = E_d(\sigma|_{Y_k}) \leq 6\omega_{\sigma|_{Y_k}}(\frac{\lambda_k}{2d})$. Recall that any modulus of continuity $\omega_f$ is subadditive (i.e. $\omega_f(x + y) \leq \omega_f(x) + \omega_f(y)$ for all $x, y$); see (Rivlin, 1981, Chap. 1). Thus for fixed $d$, we have $\omega_{\sigma|_{Y_k}}(\frac{\lambda_k}{2d}) \in \mathcal{O}(\lambda_k)$, which implies $(k \mapsto \|\sigma_k - \sigma\|_{\infty, Y_k}) \in o(\lambda_k^{1+\gamma})$. □

### C.2 PROOF OF LEMMA 5.11

Our proof of Lemma 5.11 is a straightforward application of both the Cayley–Menger determinant formula and the Leibniz determinant formula. For each $0 \leq i \leq N$, let $\widehat{S}_i(\lambda) := S(\lambda)\backslash\{p_i(\lambda)\}$, and let $\widehat{\Delta}_i(\lambda)$ be the convex hull of $\widehat{S}_i(\lambda)$. Let $\mathcal{V}(\Delta(\lambda))$ (resp. $\mathcal{V}(\widehat{\Delta}_i(\lambda))$) denote the $N$-dimensional (resp. $(N-1)$-dimensional) volume of $\Delta(\delta)$ (resp. $\widehat{\Delta}_i(\lambda)$). Define the $(N + 2)$-by-$(N + 2)$ matrix $M(\lambda) = [M_{i,j}(\lambda)]_{0 \leq i,j \leq N+1}$ as follows: $M_{i,j}(\lambda) = \|p_i(\lambda) - p_j(\lambda)\|_2^2$ for all $0 \leq i, j \leq N$; $M_{N+1,i}(\lambda) = M_{i,N+1}(\lambda) = 1$ for all $0 \leq i \leq N$; and $M_{N+1,N+1}(\lambda) = 0$.

The Cayley–Menger determinant formula gives $[\mathcal{V}(\Delta(\lambda))]^2 = \frac{(-1)^{N+1}}{(N!)^2 2^N} \det(M(\lambda))$. Analogously, if we let $M'(\lambda)$ be the square submatrix of $M(\lambda)$ obtained by deleting the first row and column from

$M(\lambda)$, then $[\mathcal{V}(\widehat{\Delta}_0(\lambda))]^2 = \frac{(-1)^N}{((N-1)!)^2 2^{N-1}} \det(M'(\lambda))$. Now, $\mathcal{V}(\Delta(\lambda)) = \frac{1}{N}\mathcal{V}(\widehat{\Delta}_0(\lambda))h_0(\lambda)$, so

$$[h_0(\lambda)]^2 = \frac{-1}{2N}\frac{\det(M(\lambda))}{\det M'(\lambda)}. \tag{11}$$

Without loss of generality, assume that $r_0 \geq r_1 \geq \dots$. Also, for any integer $k \geq 0$, let $\mathfrak{S}_k$ be the set of all permutations on $\{0, \dots, k\}$, and let $\mathfrak{S}'_k$ be the subset of $\mathfrak{S}_k$ consisting of all permutations that are not derangements. (Recall that $\tau \in \mathfrak{S}_k$ is called a *derangement* if $\tau(i) \neq i$ for all $0 \leq i \leq k$.) The diagonal entries of $M(\lambda)$ are all zeros, so by the Leibniz determinant formula, we get

$$\det(M(\lambda)) = \sum_{\tau \in \mathfrak{S}'_{N+1}} \mathrm{sgn}(\tau) \prod_{0 \leq i \leq N+1} M_{i,\tau(i)}(\lambda),$$

where $\mathrm{sgn}(\tau)$ denotes the sign of the permutation $\tau$. Note that $M_{i,j}(\lambda) \in \Theta(\lambda^{2\max\{r_i,r_j\}})$ for all $0 \leq i, j \leq N$ satisfying $i \neq j$. (Here, $\Theta$ refers to $\Theta$-complexity.) Consequently, using the fact that $M_{i,N+1}(\lambda) = M_{N+1,i} = 1$ for all $0 \leq i \leq N$, we get that $\det(M(\lambda)) \in \Theta(\lambda^{2R_N})$, where

$$R_N = \begin{cases} 2r_0 + \dots + 2r_{(N-2)/2} = 2\sum_{t=0}^{(N-2)/2} r_t, & \text{if } N \text{ is even;} \\ 2r_0 + \dots + 2r_{(N-3)/2} + r_{(N-1)/2} = -r_{(N-1)/2} + \sum_{t=0}^{(N-1)/2} r_t; & \text{if } N \text{ is odd.} \end{cases}$$

The even case corresponds to the derangement $\tau \in \mathfrak{S}_{N+1}$ given by $\tau(i) = N - i$ for $0 \leq i \leq \frac{N-2}{2}$, $\tau(\frac{N}{2}) = N+1$, $\tau(N+1) = \frac{N}{2}$; while the odd case corresponds to the derangement $\tau \in \mathfrak{S}_{N+1}$ given by $\tau(i) = N - i$ for $0 \leq i \leq \frac{N-3}{2}$, $\tau(\frac{N-1}{2}) = \frac{N+1}{2}$, $\tau(\frac{N+1}{2}) = N+1$, $\tau(N+1) = \frac{N-1}{2}$. A formula for $\det(M'(\lambda))$ can be analogously computed. Consequently, it follows from (11) that $[h_0(\lambda)]^2 \in \Theta(\lambda^{2[2r_0 - r_{\lfloor N/2 \rfloor}]})$. Now, $r_0 \geq r_{\lfloor N/2 \rfloor}$ by assumption, and $r_0$ (being the largest) must satisfy $r_0 > r_{\min}$, thus $h_0(\lambda) \in \Omega(\lambda^{r_0})$, and the assertion follows by taking $\gamma = r_0 - r_{\min}$. $\qquad\square$

### C.3 Proof of Lemma 5.12

Consider any open neighborhood $U$ of $\mathbf{0}_M$. Since $\varphi$ is open and $\varphi(\mathbf{0}_M) = \mathbf{0}_N$, the image $\varphi(U)$ must contain an open neighborhood of $\mathbf{0}_N$. Thus for any $\varepsilon > 0$, we can always choose $N+1$ points $w_0, \dots, w_N$ in $B_\varepsilon^M \setminus \{\mathbf{0}_M\}$, such that the convex hull of $\{\varphi(w_0), \dots, \varphi(w_N)\}$ contains the point $\mathbf{0}_N$. Since $\varphi(\lambda x) \geq \lambda\varphi(x)$ for all $x \in \mathbb{R}^M$, $\lambda > 0$, and since $\varphi$ is continuous, it then follows from definition that for every $k \in \mathbb{N}$, we can choose $N+1$ points $u_0^{(k)}, \dots, u_N^{(k)}$ in $U_k$, such that the convex hull of $U'_k := \{\varphi(u_0^{(k)}), \dots, \varphi(u_N^{(k)})\}$ contains $\mathbf{0}_N$. Define $r_k := \sup\{r > 0 : B_r^N \subseteq \varphi(B_{\lambda_k}^m)\}$ for each $k \in \mathbb{N}$, and note also that $\lim_{k\to\infty} r_k = \infty$. Thus, given a ball $B_r^N$ of any desired radius, there is some (sufficiently large) $k$ such that the convex hull of $U'_k$ contains $B_r^N$.

Now, since $\theta\lambda_k < \|u_j^{(k)}\|_2 \leq \lambda_k$ and $\varphi(\lambda u_j^{(k)}) \geq \lambda\varphi(u_j^{(k)})$ for all $0 \leq j \leq N$, $\lambda > 0$, we infer that none of the points $\varphi(u_0^{(k)}), \dots, \varphi(u_N^{(k)})$ are contained in the ball $B_{\theta r_k}^N$. Consequently, as $k \to \infty$, we have $\theta r_k \to \infty$, and therefore the barycentric coordinate vector $(b_0, \dots, b_N)$ (w.r.t. $U'_k$) of every point in the fixed ball $B_\tau^N$ would converge to $(\frac{1}{N}, \dots, \frac{1}{N})$ (which is the barycentric coordinate vector of the barycenter w.r.t. $U'_k$); this proves our assertion. $\qquad\square$

## D Conjectured optimality of upper bound $\mathcal{O}(\varepsilon^{-n})$ in Theorem 3.2

It was conjectured by Mhaskar (1996) that there exists some smooth non-polynomial function $\sigma$, such that at least $\Omega(\varepsilon^{-n})$ hidden units is required to uniformly approximate every function in the class $\mathfrak{S}$ of $C^1$ functions with bounded Sobolev norm. As evidence that this conjecture is true, a heuristic argument was provided in (Mhaskar, 1996), which uses a result by DeVore et al. (1989); cf. (Pinkus, 1999, Thm. 6.5). To the best of our knowledge, this conjecture remains open. If this conjecture is indeed true, then our upper bound $\mathcal{O}(\varepsilon^{-n})$ in Theorem 3.2 is optimal for general continuous non-polynomial activation functions.

For specific activation functions, such as the logistic sigmoid function, or any polynomial spline function of fixed degree with finitely many knots (e.g. the ReLU function), it is known that the minimum number $N$ of hidden units required to uniformly approximate every function in $\mathfrak{S}$ must satisfy $(N \log N) \in \Omega(\varepsilon^{-n})$ (Maiorov & Meir, 2000); cf. (Pinkus, 1999, Thm. 6.7). Hence there is still a gap between the lower and upper bounds for $N$ in these specific cases. It would be interesting to find optimal bounds for these cases.

