# OpenReview forum: "A closer look at the approximation capabilities of neural networks"
_ICLR.cc/2020/Conference — Accept (Poster)_

### Official Review · AnonReviewer2 · 2019-10-23
**Official Blind Review #2**

**Rating:** 6

**Review:**

The authors derive the universal approximation property proofs algebraically. They note that this holds even with very strong constraints on the non-bias weights.

They assert that their results are general to other kinds of neural networks and similar learners. They leave the paper with a question regarding limitations on bias weights.

I do not feel qualified to review this paper. I have opted for a weak accept since it seems thorough and the conclusions offer promise for other applications. However, I will defer to other, more qualified reviewers who have more carefully reviewed the paper than I have.

**Experience Assessment:**

I do not know much about this area.

**Review Assessment: Checking Correctness Of Derivations And Theory:**

I did not assess the derivations or theory.

**Review Assessment: Checking Correctness Of Experiments:**

N/A

**Review Assessment: Thoroughness In Paper Reading:**

N/A

---

> ### Author Response · Authors · 2019-11-14
> **Response to Review #2**
>
> Thank you very much for appreciating our work!
>
> We have made changes to our paper to improve clarity. Please see our responses to Reviews #3 and #4 for the details of these changes.

---

### Official Review · AnonReviewer3 · 2019-10-26
**Official Blind Review #3**

**Rating:** 6

**Review:**

This this exciting submission presents a new proof of Leshno's version of the universal approximation property (UAP) for neural networks  -- one of the foundational pillars of our understanding of neural networks. The new proof provides new insights into the universal approximation property. I consider these the main contribution of the paper. Specifically, the authors
- provide an upper bound on the required width for the neural network
- show that the approximation property still holds even if strong further requirements are imposed on the weights of the first or last layer.

I rate this submission a weak accept. It’s a very good paper. The work makes useful contributions that should and will be of interest to many in the field. The paper is generally well-written.


Some remarks:

- Being somewhat long, the “Proof of Theorem 3.1” would be a much better read if the authors prefixed it  with an outline of the strategy that the proof takes.

- The authors point out that the lack of dependence of Theorem 3.1 on epsilon is surprising, and cite Lin’s work from 2017 who previously found such an independence. Lin’s derivation of the epsilon-independent UAP is much more intuitive than that of this submission, in which the epsilon independence really pops out somewhat magically and for me only made sense when I read the paper again. I would encourage the authors to add to Lin’s paper’s citation sentence that this paper motivates the epsilon independence well. Alternatively, the authors could add a few sentences to their paper to provide intuition on how the epsilon-independence comes about in their line of argument.


**Experience Assessment:**

I have read many papers in this area.

**Review Assessment: Checking Correctness Of Derivations And Theory:**

I carefully checked the derivations and theory.

**Review Assessment: Checking Correctness Of Experiments:**

N/A

**Review Assessment: Thoroughness In Paper Reading:**

I read the paper thoroughly.

---

> ### Author Response · Authors · 2019-11-14
> **Response to Review #3**
>
> Thank you very much for appreciating our work!
>
> Following your suggestion, we have prefixed the "Proof of Theorem 3.1" with an "Outline of strategy for proving Theorem 3.1". We hope that the new outline helps improve clarity, and hopefully captures the underlying intuition of our proof. In particular, we have highlighted (at least an important part of) the underlying intuition for why our upper bound is independent of epsilon.

---

### Official Review · AnonReviewer4 · 2019-11-03
**Official Blind Review #4**

**Rating:** 6

**Review:**

This paper studies the representation power of single layer neural networks with continuous non-polynomial activation, and specifically, provided a refinement for the universal approximation theorem:
1.  Established an exact upper bound on the width needed to uniformly approximate polynomials of a finite degree (to any accuracy of which the upper bound is independent), and
2.  using this error-free bound to deduce a rate (of width) for approximating continuous functions.

The writing of the paper is concrete and solid.  The techniques used in establishing the results are interesting, in that:
1.  The proof for polynomial approximation (Thm 3.1) is direct, via a close examination of the Wronskian of the target polynomial function, and
2.  the analysis provided that the abilty to universally approximate is also preserved after placing certain restriction on the magnitude of the weights in the approximating neural network.  Consequently, this property is inherited by continuous function approximation to which the result is extended (Thm 3.2).
3.  This analysis and some of the results derived in the proof may be used for other analyses, e.g. representation power of multilayer networks.

Some further discussion of the results may be of interest to the readers.
-  (Optimality of Thm 3.2).  When the result in Thm 3.1 is extended to general continous functions via Jackson's theorem, to what extend does the rate deteriorate?  What does the rate look like when using certain common activations (such as ReLU, sigmoid).
-  (Reference to random features).  Thm 3.3 appears to be related to random feature representation, whose approximating ability has been studied in prior works.  Some comment on those results may be beneficial (e.g. https://arxiv.org/abs/1810.04374).
-  Although already a straightforward proof, it seems natural, and as a result may promote the presentation and clarity, to organize the proof to Thm 3.1 using smaller parts, which currently spans over 2 pages.

**Experience Assessment:**

I have read many papers in this area.

**Review Assessment: Checking Correctness Of Derivations And Theory:**

I assessed the sensibility of the derivations and theory.

**Review Assessment: Checking Correctness Of Experiments:**

N/A

**Review Assessment: Thoroughness In Paper Reading:**

I made a quick assessment of this paper.

---

> ### Author Response · Authors · 2019-11-14
> **Response to Review #4**
>
> Thank you very much for appreciating our work!
>
> (Optimality of Thm. 3.2) This is a great question! We believe the upper bound for $N$ in Theorem 3.2 is optimal, and we have included a new Appendix B in the revised paper to discuss this (conjectured) optimality. To put your question into context, it was conjectured by Mhaskar (1996) that there exists some smooth non-polynomial activation function such that at least $\Omega(\varepsilon^{-n})$ hidden units is required to uniformly approximate every function in the class of $C^1$ functions with bounded Sobolev norm. Mhaskar provided a heuristic argument for why this conjecture should be true. If Mhaskar's conjecture is indeed true, then our upper bound in Theorem 3.2 is optimal.
> For specific activation functions sigmoid and ReLU, it is already known that $(N \log N) \in \Omega(\varepsilon^{-n})$ for the class of $C^1$ functions with bounded Sobolev norm, so there is still a gap between the lower and upper bounds for $N$ in these specific cases. It would be interesting to find optimal bounds for these cases.
>
> (Reference to random features) Thank you for pointing this out! We have included a new paragraph in the Discussion section (Sec. 4).
>
> (clarity of proof of Thm 3.1) Following a suggestion by AnonReviewer3, we have prefixed the "Proof of Theorem 3.1" with an "Outline of strategy for proving Theorem 3.1". We hope that the new outline helps improve clarity.

---

### Official Review · AnonReviewer3 · 2019-11-23
**Official Blind Review #3**

**Rating:** 8

**Review:**

UPDATE TO MY EARLIER REVIEW
============================

Since this paper presets new findings that will be of significant interest to much of ICLR's audience, and the paper is is well-written, I am changing my rating to "Accept". Since Reviewer #1 did not submit a review and Reviewer #2 indicated that (s)he does not feel well-qualified to review this paper (it is very much on the theoretical side after all), it would be great to get one further review from an area chair or otherwise qualified person.

MY EARLIER REVIEW
=================

This this exciting submission presents a new proof of Leshno's version of the universal approximation property (UAP) for neural networks  -- one of the foundational pillars of our understanding of neural networks. The new proof provides new insights into the universal approximation property. I consider these the main contribution of the paper. Specifically, the authors
- provide an upper bound on the required width for the neural network
- show that the approximation property still holds even if strong further requirements are imposed on the weights of the first or last layer.

I rate this submission a weak accept. It’s a very good paper. The work makes useful contributions that should and will be of interest to many in the field. The paper is generally well-written.


Some remarks:

- Being somewhat long, the “Proof of Theorem 3.1” would be a much better read if the authors prefixed it  with an outline of the strategy that the proof takes.

- The authors point out that the lack of dependence of Theorem 3.1 on epsilon is surprising, and cite Lin’s work from 2017 who previously found such an independence. Lin’s derivation of the epsilon-independent UAP is much more intuitive than that of this submission, in which the epsilon independence really pops out somewhat magically and for me only made sense when I read the paper again. I would encourage the authors to add to Lin’s paper’s citation sentence that this paper motivates the epsilon independence well. Alternatively, the authors could add a few sentences to their paper to provide intuition on how the epsilon-independence comes about in their line of argument.

**Experience Assessment:**

I have read many papers in this area.

**Review Assessment: Checking Correctness Of Derivations And Theory:**

I carefully checked the derivations and theory.

**Review Assessment: Checking Correctness Of Experiments:**

N/A

**Review Assessment: Thoroughness In Paper Reading:**

I read the paper thoroughly.

---

### Decision · Program_Chairs · 2019-12-19

**Decision:**

Accept (Poster)

**Comment:**

This is a nice paper on the classical problem of universal approximation, but giving a direct proof with good approximation rates, and providing many refinements and ties to the literature.

If possible, I urge the authors to revise the paper further for camera ready; there are various technical oversights (e.g., 1/lambda should appear in the approximation rates in theorem 3.1), and the proof of theorem 3.1 is an uninterrupted 2.5 page block (splitting it into lemmas would make it cleaner, and also those lemmas could be useful to other authors).

---

> ### Author Response · Authors · 2020-02-14
> **Lemmas extracted from proof of Theorem 3.1, and technical oversights corrected**
>
> Thank you very much for appreciating our work! Based on your suggestion, we have extracted several lemmas from the proof of Theorem 3.1. These lemmas have been formulated to be more general than what is required in the proof of Theorem 3.1, so that they could (hopefully) be useful to other authors. We have also carefully checked through the paper; all typos have been fixed, and there were a couple of technical oversights that were spotted and duly corrected. Note that the statement of Theorem 3.1 is correct, whether or not the second instance of lambda in (i) is replaced by 1/lambda. Overall, the strategy for proving Theorem 3.1 remains the same, but we have reorganized the proof to improve clarity.